# Mammalian brain glycoproteins exhibit diminished glycan complexity compared to other tissues

Sarah E. Williams[1,2,7], Maxence Noel [2], Sylvain Lehoux[2], Murat Cetinbas[3], Ramnik J. Xavier [3,4], Ruslan I. Sadreyev[3], Edward M. Scolnick[1,5], Jordan W. Smoller [1,5,6], Richard D. Cummings [2,8] & Robert G. Mealer [1,2,5,6,8✉]

Glycosylation is essential to brain development and function, but prior studies have often been limited to a single analytical technique and excluded region- and sex-specific analyses. Here, using several methodologies, we analyze Asn-linked and Ser/Thr/Tyr-linked protein glycosylation between brain regions and sexes in mice. Brain N-glycans are less complex in sequence and variety compared to other tissues, consisting predominantly of high-mannose and fucosylated/bisected structures. Most brain O-glycans are unbranched, sialylated O-GalNAc and O-mannose structures. A consistent pattern is observed between regions, and sex differences are minimal compared to those in plasma. Brain glycans correlate with RNA expression of their synthetic enzymes, and analysis of glycosylation genes in humans show a global downregulation in the brain compared to other tissues. We hypothesize that this restricted repertoire of protein glycans arises from their tight regulation in the brain. These results provide a roadmap for future studies of glycosylation in neurodevelopment and disease.

[1] Psychiatric and Neurodevelopmental Genetics Unit, Massachusetts General Hospital, Harvard Medical School, Boston, MA, USA. [2] Department of Surgery, Beth Israel Deaconess Medical Center, Harvard Medical School, Boston, MA, USA. [3] Department of Molecular Biology, Massachusetts General Hospital, Harvard Medical School, Boston, MA, USA. [4] Center for Computational and Integrative Biology, Massachusetts General Hospital, Harvard Medical School, Boston, MA, USA. [5] The Stanley Center for Psychiatric Research at Broad Institute of Harvard/MIT, Cambridge, MA, USA. [6] Center for Precision Psychiatry, Department of Psychiatry, Massachusetts General Hospital, Harvard Medical School, Boston, MA, USA. [7] Present address: Nash Family Department of Neuroscience, Icahn School of Medicine at Mount Sinai, New York, NY, USA. [8] These authors jointly supervised this work: Richard D. Cummings, Robert G. Mealer. ✉email: rmealer@partners.org

Glycosylation regulates nearly all cellular processes and is particularly important in the development and function of the nervous system[1,2]. Glycans have been shown to influence neurite outgrowth[3], axon guidance[4], synaptogenesis[5], membrane excitability[6–9], and neurotransmission[10,11] by modulating the structure, stability, localization, and interaction properties of numerous neuronal proteins. Glycans may consist of a single monosaccharide or can be extended into elaborate sugar oligo/polysaccharides[12]. These structures are covalently attached to lipids or certain amino acids of proteins, which designates protein glycans as either N-glycans or O-glycans. Over 300 enzymes work in an elaborate assembly line to generate, attach, and modify these carbohydrate polymers, creating an immense diversity of glycan structures[2,13,14].

Despite its complexity, glycosylation is highly regulated; mutations in a single glyco-gene can lead to profound clinical syndromes, collectively termed congenital disorders of glycosylation (CDGs)[15]. The majority of CDGs present with neurologic symptoms including intellectual disability, seizures, and structural abnormalities, illustrating the particular importance of glycosylation in the brain[16]. Fine-tuning of the glycosylation pathway can also affect neurophysiology and behavior, as illustrated by the association of several glycosylation enzymes with complex human phenotypes such as schizophrenia[17,18] and intelligence[19,20].

Prior studies of brain glycosylation have typically focused on a single gene, pathway, epitope, or carrier of interest, providing insight into the roles of specific modifications. Glycolipids have been studied extensively, as they comprise the majority of glycan mass in the brain and are crucial for axon myelination, neuronal survival, and regeneration[21–23]. Proteoglycans, composed of a core protein modified by various glycosaminoglycan (GAG) chains, have also been a focus, and are known to be temporally and spatially regulated throughout brain development, serving as guidance cues during cell migration and axon pathfinding[24–26]. Less attention has been paid to N- and O-linked protein glycosylation, with a few studies showing the importance of particular modifications such as the Lewis X antigen (Le^X)[3,27–29], human natural killer antigen (HNK-1)[30,31], polysialic acid[32,33], bisecting GlcNAc[34,35], and O-mannosylation[36–38].

Systematic approaches to capture the diversity of all protein glycans in the brain have been attempted using glycomic analysis[37,39–44], glycoproteomics[45–48], microarrays[49], western blotting[50], and MALDI-Imaging techniques[51,52]. While the majority of these have produced complementary results, they tend to be individually limited by sample size, regional specificity, a single sex, or the technical constraints of a single method. For a more complete picture of brain protein glycosylation, we analyzed the frontal cortex, hippocampus, striatum, and cerebellum of male and female C57BL/6 mice using multiple validated techniques, and present a comprehensive portrait of N- and O-glycosylation in the brain characterized by a surprisingly restricted set of glycans and overall downregulation of the pathway.

## Results

**Protein N-glycosylation shows a unique but consistent pattern across brain regions.** We analyzed protein glycosylation across multiple brain regions in parallel using MALDI-TOF mass spectrometry (MS), tandem mass spectrometry (MS/MS), lectin western blotting, and RNA sequencing, with a goal of four samples per group for quantitative studies (Fig. 1A). In male mice and a commercially available human sample, the permethylated N-glycome of cortex contains a predominance of low molecular weight N-glycans (<2500 $m/z$), in striking contrast to other well studied tissues such as plasma, which are dominated by larger

(>2500 $m/z$) structures (Fig. 1B). Representative MALDI spectra from the cortex, hippocampus, striatum, and cerebellum show a similar pattern of N-glycans across each brain region (Fig. 2A). Overall, 95 unique N-glycan masses above our signal/noise cutoff were annotated across the four regions (Supplementary Data 1). Masses corresponding to multiple glycan isomers (shown in brackets), were analyzed by MS/MS as described below to confirm that each distinct isomer shown contributed to the observed signal. Most tissue N-glycomes are dominated by complex, branched N-glycans terminating with galactose and sialic acid. In contrast, the bulk of the brain N-glycome was comprised of high-mannose structures containing the two core GlcNAc and five to nine mannose residues (Fig. 2B), which are often considered proximal precursors along the synthetic pathway and found at low abundance in most tissues[53]. Five of the top 10 most abundant N-glycans in the brain were high-mannose structures, including the most abundant, $Man_5GlcNAc_2$ (Man-5), which comprised nearly half of the total glycan signal in the brain (Fig. 2B).

For further analysis, individual glycans were categorized by monosaccharide composition or shared structural characteristics such as branching (Supplementary Note 1, Supplementary Data 2), and the abundance of these groups were compared between regions. Consistently across the brain, N-glycans were predominantly high-mannose (~60%), fucosylated (~35%), and bisected (~30%) structures (Table 1). We noted a low abundance of galactose containing N-glycans (10–15%) and an even smaller amount containing sialic acid (1–3%). Of the few sialylated N-glycans detected in the brain, all were modified by the N-acetylneuraminic acid (NeuAc) form of the sugar and not the N-glycolylneuraminic acid (NeuGc), consistent with prior studies and the lack of expression of the enzyme which converts NeuAc to NeuGc in the brain[54]. Further, the lack of NeuGc detected in the brain supports minimal contribution from blood to the observed signal, given that the dominant N-glycans in murine blood are disialylated NeuGc structures[54,55].

The cerebellum was the most unique of the four brain regions analyzed. Nine of the top 10 most abundant N-glycans differed between the cerebellum and other regions, including the most abundant N-glycan, Man-5 (Fig. 2B). The cerebellum also displayed significantly less paucimannose and mono-antennary structures, and a greater abundance of complex, multi-antennary, and hybrid glycans (Table 1, Fig. 2C). The cortex and hippocampus appeared most similar in their composition of N-glycans, and the trend toward less complex and branched structures compared to the cerebellum (Table 1, Fig. 2C). Tandem MS data confirming our structural assignments of the most abundant N-glycans, as well as a description of the classification of different glycan categories, is included in the supplementary material (Supplementary Note 2, Supplementary Fig. 1).

**Endo H treatment confirms the predominance of high-mannose and hybrid N-glycans in the brain.** Given the surprising abundance of high-mannose N-glycans identified in the brain by MALDI-MS, we sought to further confirm this observation using an enzyme that cleaves only high-mannose and hybrid structures, known as endoglycosidase H (Endo H). The intensity of individual N-glycans isolated from the cortex using PNGase F (Fig. 3A) was compared to those isolated by Endo H (Fig. 3B) and those from a subsequent PNGase F digestion following Endo H treatment (Fig. 3C). This allowed for the discrimination of structures that are Endo H sensitive, such as high-mannose and hybrid species, and those that are Endo H insensitive, such as pauci-mannose and complex N-glycans. Of note, PNGase F and Endo H

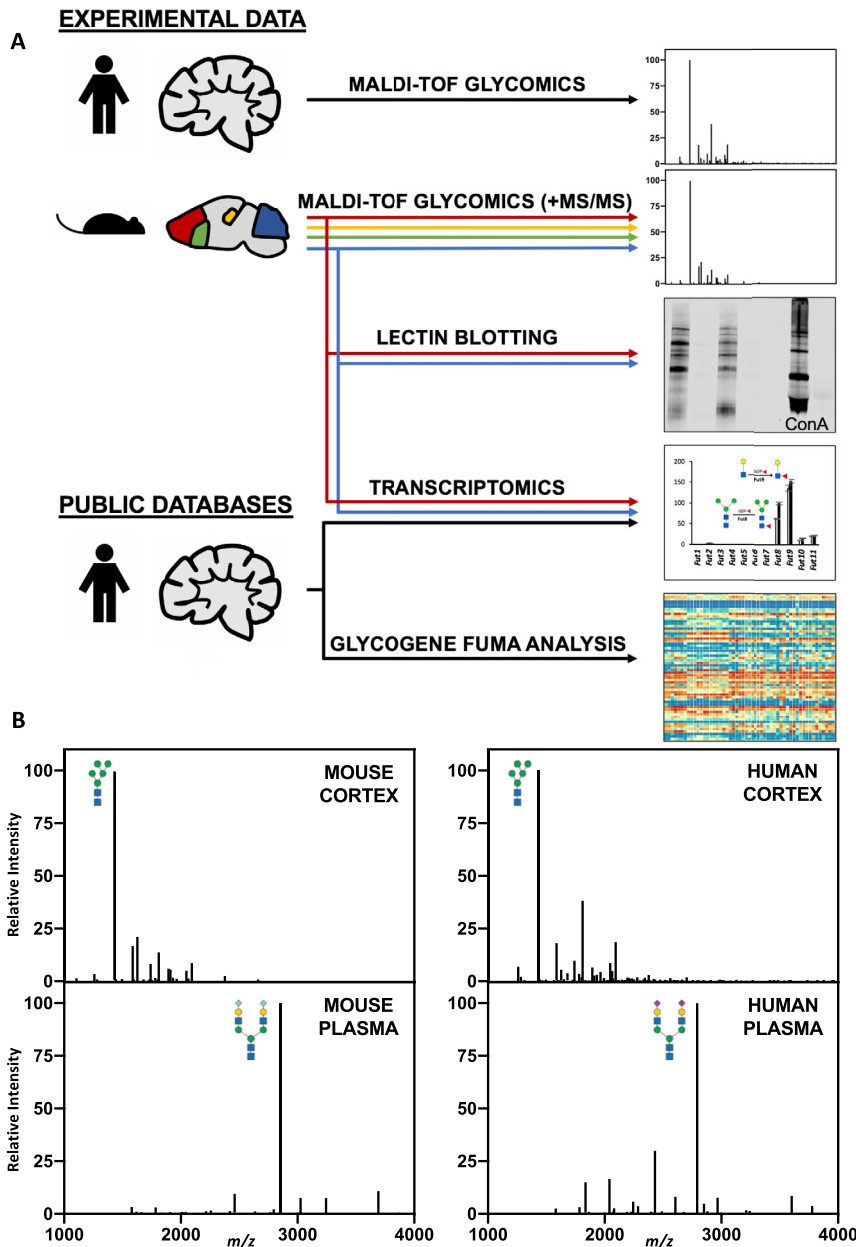

**Fig. 1 The brain N-glycome was distinct in its restricted glycan repertoire. A** Schematic of workflow analyzing the brain glycome in human and mouse samples. **B** Representative MALDI MS data of permethylated N-glycans from mouse and human cortex showed a predominance of low-molecular-weight structures, both with a major peak of Man-5 at *m/z*: 1579. In contrast, N-glycans from mouse and human plasma have a primary peak of A2G2S2 (*m/z*: mouse 2852, human 2792) and more complex and high-molecular-weight structures.

have a different cleavage site on N-glycans, which results in a difference of one GlcNAc residue between the two digestions and prevents the discernment of structures with and without a core fucose following Endo H treatment. As such, we focused our comparison on the abundance of PNGase F-released glycans before and after Endo H treatment (Fig. 3A vs Fig. 3C) to determine Endo H sensitivity of each parent peak.

Endo H effectively removed 100% of the high-mannose structures present on glycoproteins in the cortex, as none were detected after subsequent PNGase F treatment (Fig. 3C). Structures corresponding to Man-5-9 were detected in the Endo H spectra, further supporting this conclusion (Fig. 3B). On the contrary, known complex and paucimannose N-glycans were not sensitive to Endo H treatment; these glycans were present at the same relative intensity after the secondary PNGase F treatment

(Fig. 3C), and no structures corresponding to these glycans were detected in the Endo H spectra (Fig. 3B).

Several of the top 15 N-glycan masses identified in the brain had potentially ambiguous structures, as their composition of monosaccharides could form either a hybrid or complex N-glycan. For example, the MS peak at *m/z*: 2070 (HexNA-c4Hex5) could represent a common plasma N-glycan with two antenna and two terminal galactose residues (A2G2), or a bisected hybrid glycan lacking terminal galactose (A1BH5). Endo H digestion revealed that the N-glycan at *m/z*: 2070 is predominantly the hybrid species A1BH5, as its corresponding mass was detected in the Endo H MALDI spectra (Fig. 3B) with minimal signal in the PNGase F spectra after Endo H treatment (Fig. 3C). In contrast, another potentially ambiguous glycan (*m/z*: 2214, denoted as F2A2G1, F2A1G1B) was completely insensitive

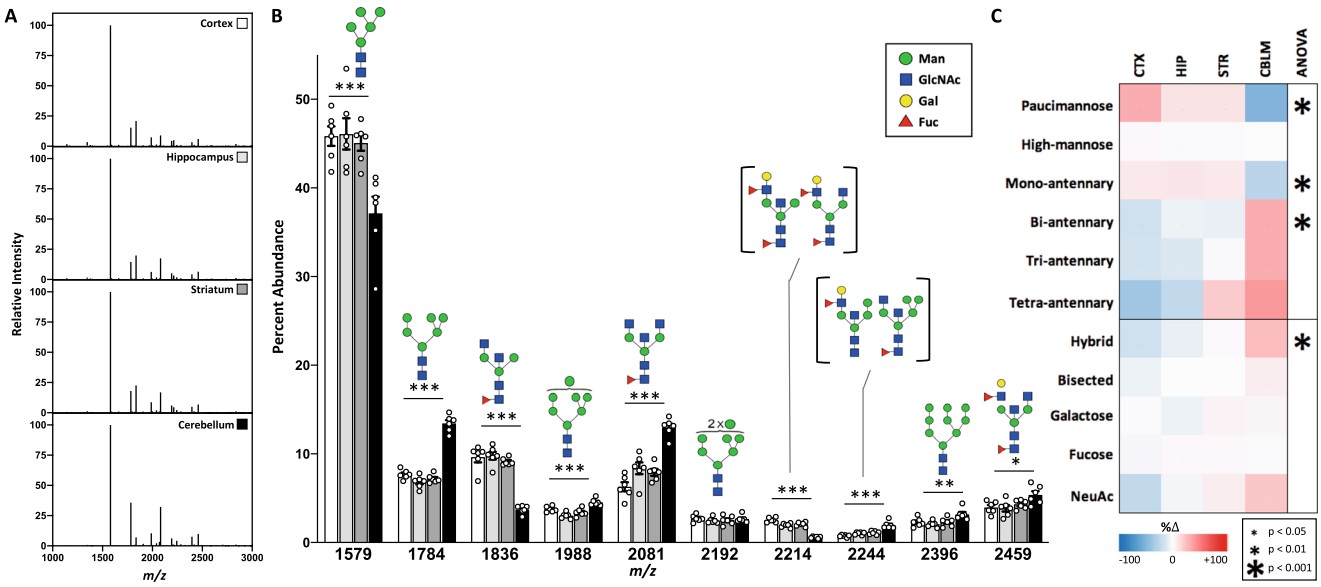

**Fig. 2 Protein N-glycomics revealed an abundance of high-mannose and fucosylated/bisected structures across the mouse brain. A** Representative MALDI spectra of protein N-glycans isolated from four brain regions show a similar overall pattern. **B** The 10 most abundant N-glycan masses differ between brain regions ($n = 6$ per region, male mice). Data presented as mean percent abundance $+/-$ SEM for the 10 most abundant N-glycan masses averaged across regions. Corresponding glycan structures are presented above each peak, with distinct isomers corresponding to the same mass shown in brackets. For single-factor ANOVA calculations, $df = (3,20)$, $F_{crit} = 3.098$, p values *<0.05, **<0.01, and ***<0.001. **C**) Categorical analysis of N-glycans demonstrates greater abundance of complex structures in the cerebellum, with a heat map showing percent change from the average of four regions. CTX cortex, HIP hippocampus, STR striatum, CBLM cerebellum. Source data are provided as a Source Data file.

**Table 1 Brain N-glycans.**

| Glycan Category | Cortex | Hippocampus | Striatum | Cerebellum | F-value | p-value |
|---|---|---|---|---|---|---|
| Paucimannose | 3.1 ± 0.2 | 2.6 ± 0.3 | 2.6 ± 0.3 | 1.2 ± 0.1 | **13.23** | **0.00005** |
| High-mannose | 62.6 ± 1.3 | 60.8 ± 1.8 | 60.8 ± 0.6 | 61.1 ± 1.8 | 0.32 | 0.81 |
| Mono-antennary | 19.0 ± 0.8 | 19.2 ± 0.7 | 18.9 ± 0.4 | 13.0 ± 0.6 | **23.39** | **0.000001** |
| Bi-antennary | 13.6 ± 0.9 | 15.5 ± 1.2 | 15.4 ± 0.3 | 21.7 ± 0.9 | **15.68** | **0.00002** |
| Tri-antennary | 1.7 ± 0.2 | 1.7 ± 0.3 | 1.9 ± 0.2 | 2.6 ± 0.4 | 2.46 | 0.09 |
| Tetra-antennary | 0.14 ± 0.04 | 0.18 ± 0.05 | 0.28 ± 0.07 | 0.32 ± 0.08 | 1.69 | 0.2 |
| Hybrid | 5.5 ± 0.2 | 6.3 ± 0.2 | 6.7 ± 0.4 | 8.3 ± 0.4 | **13.34** | **0.00005** |
| Bisected | 30.3 ± 1.1 | 32.1 ± 1.5 | 32.3 ± 0.5 | 34.4 ± 1.5 | 1.31 | 0.3 |
| Galactose | 13.5 ± 0.6 | 12.8 ± 1.2 | 14.1 ± 0.6 | 14.1 ± 1.3 | 0.4 | 0.76 |
| Fucose | 34.7 ± 1.3 | 36.2 ± 1.9 | 36.3 ± 0.6 | 35.3 ± 1.8 | 0.29 | 0.83 |
| NeuAc | 1.9 ± 0.5 | 2.3 ± 0.5 | 2.5 ± 0.6 | 2.9 ± 0.3 | 0.8 | 0.51 |
| NeuGc | 0 | 0 | 0 | 0 | – | – |

Categorical analysis of N-glycans highlighted regional differences in the brain. Data from male mice, presented as mean percent abundance $+/-$ SEM for each category. Significant F- and p-values are indicated in bold, $df = (3,20)$, $F_{crit} = 3.098$, $n = 6$ per region. Source data are provided as a Source Data file.

to Endo H digestion, indicating that glycans at this mass do not include a hybrid species, which was further supported by our MS/MS results (Supplementary Fig. 1).

In a third unique case, the peak at $m/z$: 2040 was partially Endo H sensitive, indicating a mixture of hybrid and non-hybrid glycans present at this mass. The structure corresponding to the parent hybrid glycan FA1BH4 was detected in the Endo H spectra (A1BH4, Fig. 3B) but a small amount of glycan was present in the secondary PNGase F spectra (Fig. 3C). MS/MS analysis confirmed the presence of both a hybrid structure and a complex, branched structure present at $m/z$: 2040, which explains why the signal intensity at this mass decreased after Endo H treatment but was not removed entirely (Supplementary Fig. 2).

**Brain O-glycans are primarily sialylated O-GalNAc structures.** After removal of N-glycans, we analyzed permethylated O-glycans isolated from the remaining brain glycopeptides via β-elimination

using MALDI-TOF MS. Of note, we initiated our investigation of brain O-glycans with the same number of samples for N-glycans, but no detectable signal was present for some purifications, likely resulting from a combination of their relative low absolute abundance in the brain compared to N-glycans[56] and the additional complexity of their isolation and purification (Supplementary Note 3). As such, quantitative comparisons of O-glycans were limited to groups with at least three samples. We identified 26 unique O-glycans in at least one brain region above our signal to noise threshold, which included both O-GalNAc and O-mannose (O-Man) structures (Supplementary Data 1). Representative MALDI spectra from the cortex, hippocampus, striatum, and cerebellum showed an overall similar O-glycan pattern (Fig. 4A). There were several differences in the abundance of individual O-glycans between brain regions, including the most abundant structure, a di-sialylated core 1 O-GalNAc glycan at $m/z$: 1257 and the most abundant O-Man glycan, found at $m/z$: 1100 (Fig. 4B).

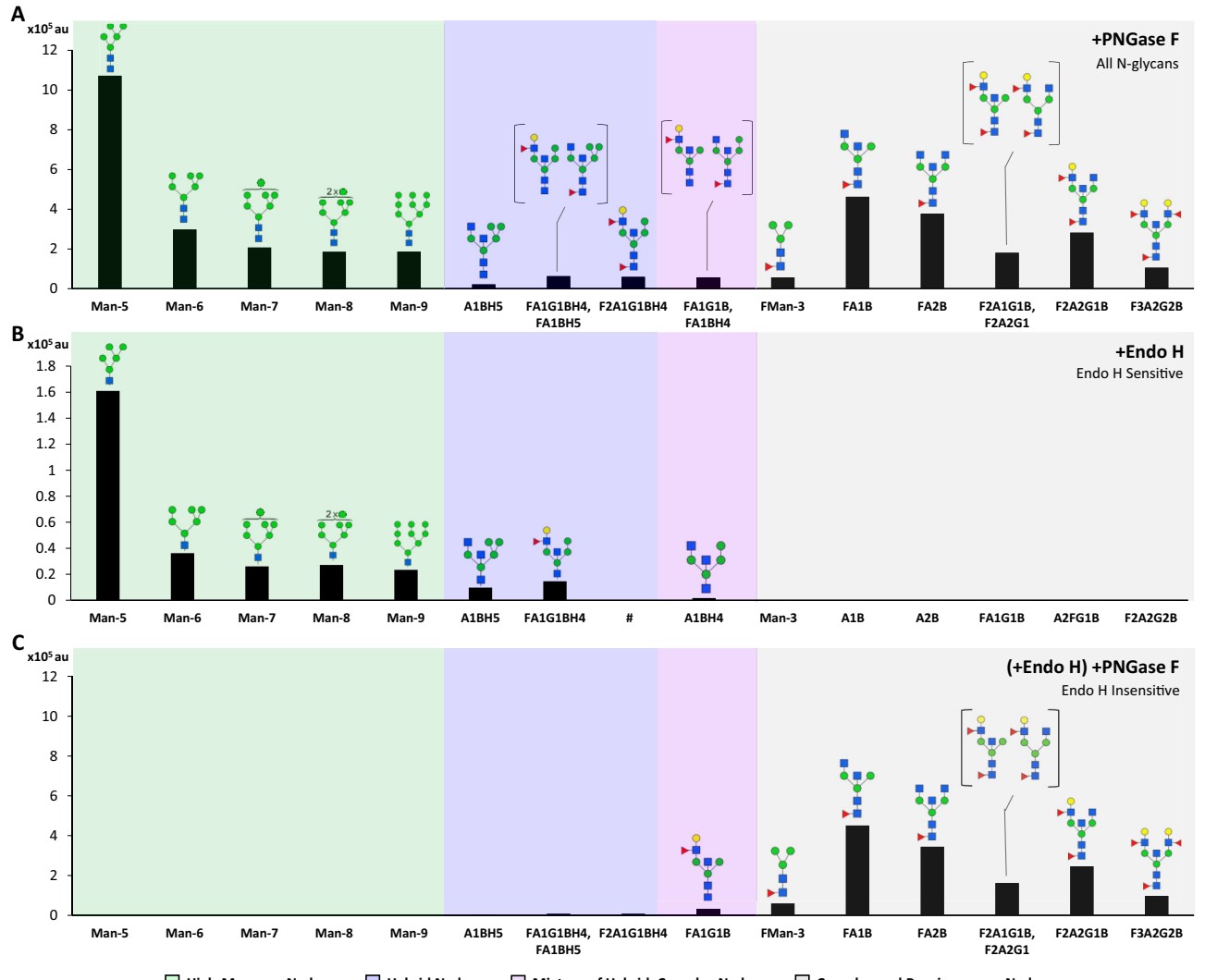

**Fig. 3 Endo H treatment distinguished high-mannose and hybrid structures from complex and paucimannose N-glycans.** The intensity of the 15 most abundant N-glycan masses from a representative male mouse cortex sample is shown after treatment with different glycosidases, grouped by high-mannose (green), hybrid (blue), mixed (pink), or complex and paucimannose structures (gray), with glycan names shown below (F - fucose; G - galactose; S - sialic acid; A - antenna; B - bisected; and H - hybrid). The y-axis (arbitrary units; au) represents the signal intensity (arbitrary units; au) for each mass measured directly from the MALDI without normalization and scaled such that Man-5 (**A** and **B**) and FA1B (**A** and **C**) were equal for visual comparison. Distinct isomers corresponding to the same mass are shown in brackets. **A** Treatment with PNGase F removed all N-glycans. **B** Treatment with Endo H removed only high-mannose and hybrid N-glycans. A placeholder (#) is included because the Endo H fragment corresponding to F2A1G1BH4 is indistinguishable from that of the same glycan without a core fucose (F1A1G1BH4), due to Endo H cleavage between the two core GlcNAc residues and proximal to the core fucose. This results in only one structure present in the +Endo H spectra corresponding to two unique parent glycans. **C**) PNGase F treatment after Endo H removed complex and paucimannose N-glycans, which were insensitive to Endo H treatment.

Analysis of all protein O-glycans stratified by structural components (Supplementary Table 1) revealed that the majority are O-GalNAc-type, comprising 74–84% of the total O-glycan signal across the brain (Table 2). The abundance of O-Man species varied significantly between brain regions, ranging from 11% of all O-glycans in the cortex, to 25% in the cerebellum (Table 2, Fig. 4C). In contrast to brain N-glycans, which had a large amount of fucose (~30%) and a paucity of sialic acid (~2%), few brain O-glycans were fucosylated (~10%), while the majority were sialylated (~90%). We noted very few O-glycans containing both sialic acid and fucose in the brain (<2% in all regions), and simple linear regression of fucosylated vs sialylated O-glycans showed a strong and highly significant negative correlation in both O-GalNAc and O-Man glycans (Supplementary Fig. 3). We

detected a small amount (1–2%) of O-glycans containing the NeuGc form of sialic acid, consistent with prior studies[37,57]. Among the dominant O-glycans detected, all of the sialylated species contain solely NeuAc (Fig. 4B). The most common O-glycan structure, *m/z*: 1257, comprises 64% of the total O-glycan abundance and contains two NeuAc residues, while the same structure containing either one or two NeuGc residues (*m/z*: 1287 and 1317) was detected at only 0.8% and 0.2% abundance, respectively (Supplementary Data 1, Supplementary Table 1). The nearly 80-fold difference between NeuAc and NeuGc abundance on brain O-glycans is again consistent with prior studies[54], as well as the minimal contribution from blood elements to the signal. The small amount of NeuGc present on brain O-glycans is presumably peripherally synthesized and recycled in the brain.

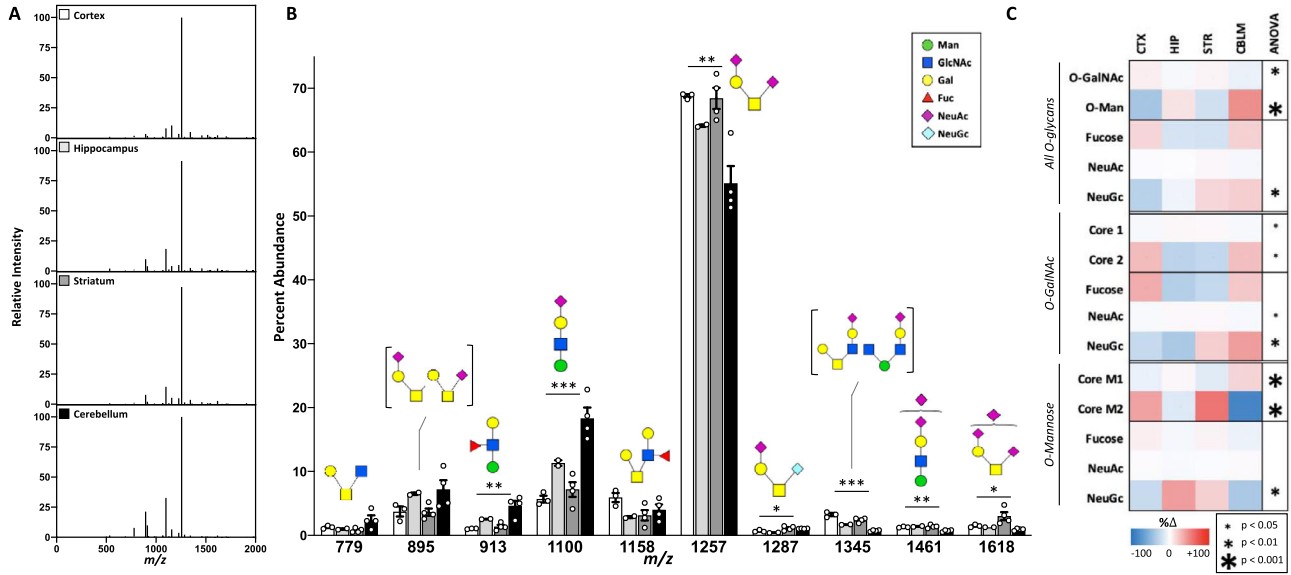

**Fig. 4 Protein O-glycomics revealed a higher proportion of O-GalNAc glycans compared to O-Man in mouse brain. A** Representative MALDI spectra of protein O-glycans isolated from four regions show a consistent pattern across the brain. **B** The 10 most abundant O-glycan masses in the brain include both O-GalNAc and O-mannose (O-Man) type structures (CTX = 3, HIP = 2, STR = 4, CBLM = 4, male mice). Data presented as mean percent abundance +/− SEM. Corresponding glycan structures are presented above each peak, with distinct isomers corresponding to the same mass shown in brackets. For single-factor ANOVA calculations, $df = (3,9)$, $F_{crit} = 3.862$. $p$ values *<0.05, **<0.01, and ***<0.001. **C** Categorical analysis of O-glycans revealed differences in the abundance of O-GalNAc, O-Man, and NeuGc-containing glycans between regions. Analyzed independently, O-GalNAc and O-Man glycans varied in their proportion of core 1, core 2, and sialylated structures. Data presented as a heat map showing percent change from the average of four brain regions. Source data are provided as a Source Data file.

**Table 2 Brain O-glycans.**

| Glycan Category | Cortex | Hippocampus | Striatum | Cerebellum | F-value | p-value |
|---|---|---|---|---|---|---|
| O-GalNAc | 84.1 ± 0.3 | 77.7 ± 0.2 | 82.3 ± 0.9 | 73.5 ± 2.5 | **8.83** | **0.005** |
| O-Man | 10.7 ± 0.4 | 18.9 ± 3 | 13.9 ± 1.1 | 25.0 ± 2.5 | **14.13** | **0.0009** |
| Fucose | 9.9 ± 1.0 | 7.1 ± 0.3 | 7.0 ± 1.2 | 10.0 ± 1.2 | 1.41 | 0.3 |
| NeuAc | 89.2 ± 0.9 | 89.7 ± 0.2 | 91.4 ± 1.2 | 86.6 ± 1.8 | 2.3 | 0.15 |
| NeuGc | 1.28 ± 0.15 | 1.72 ± 0.02 | 2.09 ± 0.08 | 2.15 ± 0.15 | **9.56** | **0.004** |
| O-GalNAc Core 1 | 90.0 ± 1.0 | 94.3 ± 0.1 | 94.1 ± 1.1 | 90.1 ± 1.2 | **4.22** | **0.04** |
| O-GalNAc Core 2 | 10.0 ± 1.0 | 5.7 ± 0.1 | 5.9 ± 1.1 | 9.9 ± 1.2 | **4.22** | **0.04** |
| O-GalNAc Fucose | 7.4 ± 1.0 | 3.8 ± 0.2 | 4.2 ± 1.1 | 6.7 ± 1.4 | 1.94 | 0.19 |
| O-GalNAc NeuAc | 90.6 ± 0.9 | 94.2 ± 0.1 | 94.3 ± 1.2 | 89.9 ± 1.3 | **4.05** | **0.04** |
| O-GalNAc NeuGc | 1.2 ± 0.2 | 1.0 ± 0.1 | 1.8 ± 0.2 | 2.1 ± 0.1 | **11.23** | **0.002** |
| O-Man Core M1 | 77.2 ± 2.4 | 85.5 ± 1.4 | 74.0 ± 4.1 | 97.0 ± 0.6 | **14.9** | **0.0008** |
| O-Man Core M2 | 22.8 ± 2.4 | 14.5 ± 1.4 | 26.0 ± 4.1 | 3.0 ± 0.6 | **14.9** | **0.0008** |
| O-Man Fucose | 22.0 ± 1.1 | 20.3 ± 0.7 | 21.7 ± 1.9 | 19.5 ± 2.2 | 0.44 | 0.73 |
| O-Man NeuAc | 78.2 ± 1.3 | 76.5 ± 0.5 | 77.1 ± 1.6 | 78.6 ± 2.5 | 0.24 | 0.87 |
| O-Man NeuGc | 2.8 ± 0.6 | 5.0 ± 0.1 | 4.3 ± 0.5 | 2.3 ± 0.3 | **7.05** | **0.0098** |

Categorical analysis of O-glycans highlighted regional differences in the brain. Data from male mice, presented as mean percent abundance +/− SEM for each category. Significant F- and $p$-values are indicated in bold, $df = (3,9)$, $F_{crit} = 3.862$, $n$ for CTX = 3, HIP = 2, STR = 4, CBLM = 4. Source data are provided as a Source Data file.

Of note, not all glycans could be classified as O-GalNAc or O-Man with confidence, as some peaks correspond to monosaccharide compositions that could form either type of structure (1–5% of the total glycan signal). For example, $m/z$: 1344, included in the top 10 O-glycans (Fig. 4B), could include both O-Man and O-GalNAc species, as has been reported in a prior study[37]. To further analyze brain O-glycans, we took those that were confirmed as O-GalNAc or O-Man based on MS/MS results (Supplementary Fig. 2) or prior reports[37,58] and normalized the abundance within each O-glycan subtype to sort by structural characteristics (Table 2 and Supplementary Table 1). Further, we excluded potential structures containing the α-Gal epitope as our results do not confidently rule in its presence, and we did not

detect the transcript for its synthetic enzyme α1,3-galactosyltransferase (*Ggta1*) in the brain[59].

Analyzed separately, O-GalNAc and O-Man glycans varied in the abundance of different core structures across brain regions (Table 2, Fig. 3C). Core 2 O-GalNAc glycans, defined by the addition of GlcNAc to the GalNAc of the core 1 structure, were highest in the cortex and cerebellum. The cerebellum had the highest abundance of O-Man glycans compared to other brain regions and were predominantly core M1 structures lacking a second GlcNAc attachment to the core mannose (Table 2). Tandem MS data confirming our structural assignments of O-glycans is included in the supplementary material (Supplementary Note 4, Supplementary Fig. 3).

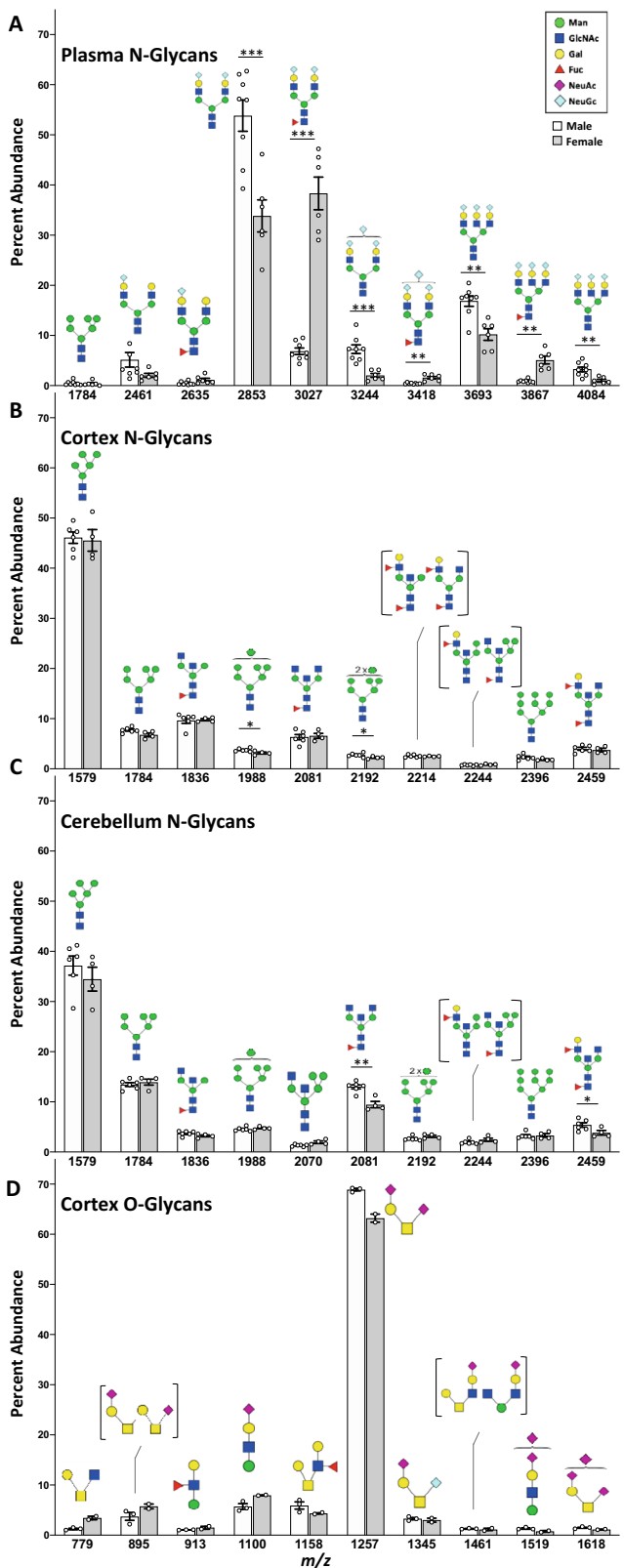

**Fig. 5 Protein glycosylation showed minimal sex differences in the brain compared to plasma in mice.** Comparison of the top 10 most abundant protein N-glycans in plasma (**A**), cortex (**B**), and cerebellum (**C**) reveals differences between sexes, with much greater divergence between male and female mice in the plasma than in either brain region. **D** Comparison of the 10 most abundant protein O-glycans in the cortex revealed minor variation between genders, though limited sample size prevented statistical analysis. For plasma samples, male = 8, female = 6. For brain N-glycans, male = 6, female = 4. For brain O-glycans, male = 3, female = 2. Corresponding glycan structures are shown above each assigned peak, with distinct isomers corresponding to the same mass shown in brackets. Data presented as mean +/− SEM for percent abundance of each peak, with unpaired two-tailed t-tests assuming unequal variance performed for sex comparisons of individual glycans. p values *<0.05, **<0.01, and ***<0.001. Source data are provided as a Source Data file.

cerebellum from male and female mice, confirming strong sex differences in the plasma but only subtle variation in the brain.

We detected 29 plasma N-glycans consisting predominantly of complex, sialylated structures modified by NeuGc sialic acid (Supplementary Data 3, Supplementary Table 2), in agreement with the previous reports[55]. There were striking sex differences in the plasma protein glycomes; the most abundant N-glycan in male mice was A2G2S2 at $m/z$: 2853, while in females the most abundant N-glycan was the fucosylated form of this same species at $m/z$: 3027 (Fig. 5A). Female mice had a 5-fold increase in all fucosylated structures compared to the male plasma glycome (Supplementary Table 3).

In the brain, sex differences in protein N-glycosylation were much less pronounced, with similar overall profiles between male and female mice in the cortex (Fig. 5B) and cerebellum (Fig. 5C) (Supplementary Table 3). The cerebellum of female mice showed less biantennary glycans, an increase in sialylation, and an overall trend toward more complex structures compared to the males. The cortex followed a similar trend but had overall less distinction between sexes. We anticipate that O-glycosylation differences exist between sexes, similar to N-glycosylation. O-glycans from the cortex of two female mice showed minor variation in individual glycan abundances compared to the males (Fig. 5D), but these data were not analyzed further due to low sample size as discussed above (Supplementary Note 3). The pattern, however, was identical to multiple female mice harboring a point mutation, which had only subtle effects on O-glycans[56], suggesting the observed O-glycan trends between sexes are consistent but not conclusive. Though not as pronounced as the differences observed in plasma, these results illustrate that brain protein glycosylation shows some sex-dependence and underscore the importance of analyzing both sexes separately.

**Lectin blotting confirms the high abundance of high-mannose, fucosylated, and bisected N-glycans in the brain**. To complement our MS findings, we performed western blotting of brain glycoproteins using several commercially available biotinylated lectins. Although lectin binding is often not specific for a single epitope, their increased affinity for certain glycan features provides important confirmatory information when used in combination with techniques such as glycomics and glycosidase sensitivity. Human plasma was included as a positive control given the abundance of literature on the human plasma N-glycome[60]. Glycoproteins were treated with or without PNGase F to determine the relative contribution of N- vs. O-glycans to the observed signal. Of note, we detected significant background binding of our fluorescent streptavidin secondary to brain glycoproteins (Supplementary Fig. 4), which likely resulted

**Sex-specific differences in protein glycosylation are minimal in the brain compared to plasma**. Previous studies of the brain glycoproteome have primarily focused on mice of a single sex[42,45,46,49,52]. However, it is known that mice show both strain and sex-specific differences in plasma protein glycosylation[55]. We compared the protein N-glycome of plasma, cortex, and

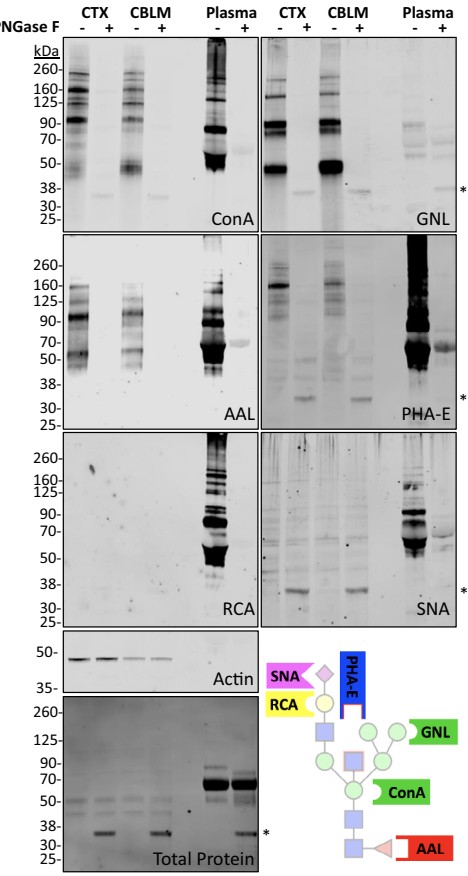

**Fig. 6 Lectin blotting supports a predominance of high-mannose, fucose, and bisected N-glycans in the brain.** Protein lysate from a representative male mouse cortex and cerebellum with human plasma as a positive control was treated with or without PNGase F and visualized using biotinylated lectins (ConA, GNL, PHA-E, AAL, RCA, and SNA) in addition to immunoblotting for actin and staining for total protein. Non-specific binding of lectins to PNGase F is noted by an asterisk (*) near 35 kDa, as shown in the Total Protein stain. Protein blotting of brain lysate with each lectin has been repeated at least three times each with similar results. A schematic with common lectin-binding sites is shown for reference. Source data are provided as a Source Data file.

the unique predominance of high-mannose N-glycans in the brain (Supplementary Fig. 5).

AAL binds fucose in both α(1–3) and α(1–6) linkages of N- and O-glycans. Strong AAL binding was observed in both brain regions and was entirely PNGase F sensitive (Fig. 6), suggesting that the bulk of fucose on glycoproteins in the brain was present on N-glycans, in agreement with our glycomics results (Table 1). PHA-E, commonly used as a marker for bisected N-glycans, showed strong binding in cortex and cerebellum samples and was PNGase F sensitive (Fig. 6). GSL-II, which recognizes terminal GlcNAc, showed a weak signal in the brain which decreased after PNGase F, consistent with the presence of terminal GlcNAc on N-glycans (Supplementary Fig. 4).

RCA binding, which recognizes galactose in both β(1–3) and β(1–4) linkages, was not detected in brain lysates, but showed a strong signal in human plasma, consistent with a relative paucity of galactose in the brain (Fig. 6), though the presence of fucose on most complex N-glycans may interfere with binding. ECL, which recognizes terminal galactose, showed weak binding in the brain that increased after treatment with the sialidase NeuA and was insensitive to PNGase F, consistent with terminal galactose on O-glycans which are commonly sialylated (Supplementary Fig. 4). WFA, which recognizes to terminal GalNAc, showed weak binding to brain lysates and was insensitive to PNGase F, further suggesting that N-glycans with the LacdiNAc motif are not abundant in the brain (Supplementary Fig. 4). SNA, also known as elderberry lectin and commonly used to detect glycans with α(2–6)-linked sialic acid, showed only trace binding that was insensitive to PNGase F, consistent with sialylation of O-glycans (Fig. 6). This finding is consistent with our glycomics data that a small minority of N-glycans contain sialic acid (~2%), whereas the majority of O-glycans (>85%) contain at least 1 sialic acid residue (Table 2), and our quantitative results showing that O-glycans are less abundant in the brain[56].

**Glycosylation gene expression correlates with glycomics and regional differences.** We next sought to determine if the expression patterns of glycosylation genes would provide insight into the unique glycome patterns observed in the brain. Comprehensive RNA sequencing and analysis was performed using the contralateral hemispheres of the cortex and cerebellum from the same male mice used in our glycomic analysis as previously described[62–64]. We generated a list of 269 known glycosyltransferases, glycosylhydrolases, sulfotransferases, and glycan-related genes based on a previous publication[20] and the Carbohydrate Active Enzymes database (CAZy)[65], after excluding genes whose transcripts were not detected in our experiment (Supplementary Data 4). A comparison between cortex and cerebellum identified 62 differentially expressed glycosylation genes, spanning all synthetic pathways, including protein N-glycans (Fig. 7A), O-GalNAc (Fig. 7B), and O-Man glycosylation (Fig. 7C).

Several correlates between the unique protein glycome and gene expression in the brain were evident. Of the N-acetylglucosaminyltransferases for N-glycans, *Mgat3* levels were much higher than those of branching *Mgat* enzymes (Fig. 7D), consistent with the high abundance of bisected N-glycans and the paucity of complex, branched N-glycans. Of the fucosyltransferases, *Fut8* and *Fut9* were most abundant (Fig. 7E), correlating with the high amount of core-fucosylated N-glycans and the Le^X antigen, respectively. Differential expression of several enzymes between cortex and cerebellum also correlated with the glycomics results. For example, the cortex shows higher expression of *Mgat5b* (Fig. 7F), the sole enzyme responsible for the synthesis of core-2 O-Man glycans[66],

from high levels of biotin-bound carboxylases in the brain relative to other tissues as previously described[61]. To reduce this non-specific binding, we pre-cleared the brain lysates by incubation and precipitation with magnetic streptavidin beads, which removed nearly all non-specific binding and allowed for sensitive detection of glycoprotein bands.

Results from lectin blotting agreed with our N-glycomics, indicating high abundances of high-mannose, fucosylated, and bisected glycans, with a near absence of galactosylated and sialylated structures (Fig. 6). ConA, which binds the core mannose structure of all N-glycans, displayed strong binding in the cortex and cerebellum which was completely sensitive to PNGase F cleavage. GNL, also known as snowdrop lectin, primarily binds extended mannose branches found in high-mannose and hybrid N-glycans. Despite minimal binding in plasma, GNL binding of glycoproteins from both brain regions was robust and PNGase F sensitive, corroborating a predominance of these structures in the brain relative to other N-glycans (Fig. 2 and Table 1). ConA binding in both brain regions was equally sensitive to PNGase F and Endo H, whereas plasma ConA binding was only slightly reduced by Endo H, further supporting

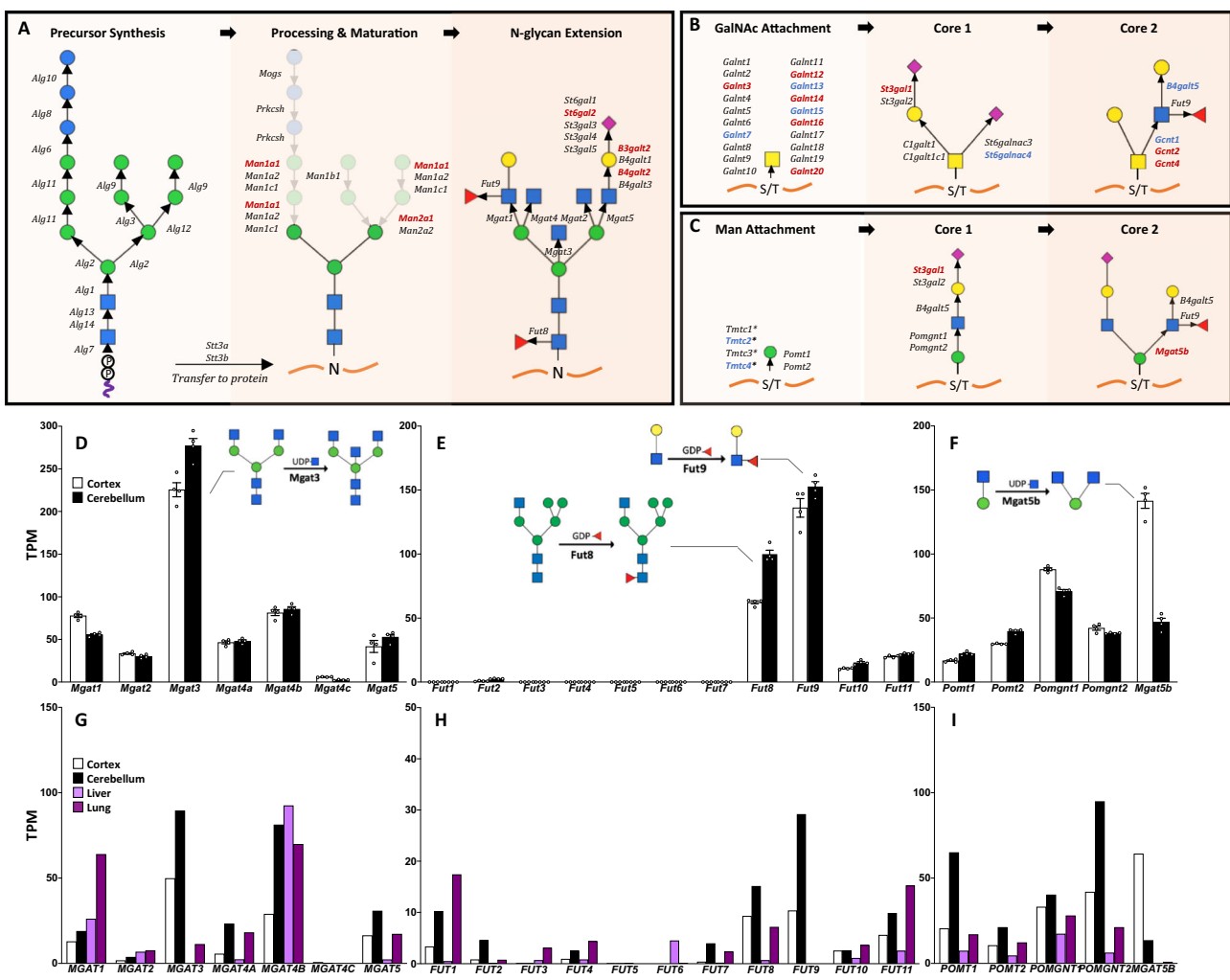

**Fig. 7 Differential RNA expression between brain regions in mice spans multiple glycosylation pathways and correlates with glycomics results.** RNA seq data from cortex and cerebellum (n = 4 each, male mice) revealed differential expression of enzymes involved in several glycosylation pathways, including the synthesis of N-glycans (**A**), O-GalNAc glycans (**B**), and O-Man glycans (**C**). Transcripts in red have significantly increased expression in the cortex relative to the cerebellum, and those in the blue are decreased, as determined using the EdgeR method with gene cutoffs of 2-fold change in expression value and false discovery rates (FDR) below 0.05. *Tmtc1-4 add O-linked Man but these residues are not extended further. Mouse brain RNA seq results for N-acetylglucosaminyltransferases (**D**), fucosyltransferases (**E**), and O-Man specific enzymes (**F**) demonstrated high RNA levels of *Mgat3*, *Fut8*, *Fut9*, and *Mgat5b*, which correlate with results from glycomics. Data for TPM presented as mean +/− SEM. For D-F, n = 4 independent samples for each brain region from different mice. Human RNA seq data from GTEx portal showed a similar expression profile for N-acetylglucosaminyltransferases (**G**), fucosyltransferases (**H**), and O-Man pathway enzymes (**I**) in the brain between humans and mice, but this pattern is distinct from human liver and lung. Source data are provided as a Source Data file.

and these structures were several-fold more abundant in this region.

**Human glycosylation genes show a global downregulation in the brain**. The unique pattern of protein glycosylation in the mouse brain is mirrored in human samples, which have a similar N-glycan MALDI profile (Fig. 1B) and show comparable abundances of high-mannose, bisected, and fucosylated glycans in prior studies[67,68]. Gene expression data of the human cortex and cerebellum downloaded from the GTEx Portal[69–71] revealed several similarities with our RNA expression data from mice for several glycosyltransferase families, including N-acetylglucosaminyltransferases (Fig. 7G), fucosyltransferases (Fig. 7H), and the enzymes of O-mannosylation (Fig. 7I). A comparison to other human tissues with well-characterized glycomes, such as liver and lung, illustrated the uniqueness of glycosylation gene expression in the brain. Both brain regions express high levels of *MGAT3* and have a high abundance

of bisected N-glycans, while lung, plasma, and liver have low levels of *MGAT3* and relatively few bisected N-glycans (Fig. 7G)[72–74]. The liver and lung have lower levels of nearly all the enzymes for O-Man synthesis (Fig. 7I), consistent with the general restricted presence of O-mannose glycans to the brain and a few other tissues[37,38,75].

Finally, we compared human glycosylation gene expression in the brain to all other tissues on a global scale. We applied the publicly available GENE2FUNC feature of the FUMA GWAS platform[76] to a list of 354 glycan-related genes in humans (Supplementary Data 5). Comparison of 54 specific tissue types revealed a distinct pattern of downregulation on the individual gene level across 13 brain regions compared to other tissues (Fig. 8A). Grouped expression analysis of 30 general tissue types showed that the brain is the only region with a significantly down-regulated gene set, and the only region which is significantly different when comparing differences in both directions (Fig. 8B). Further analysis of the 13 brain regions as independent tissues shows some regional differences, particularly

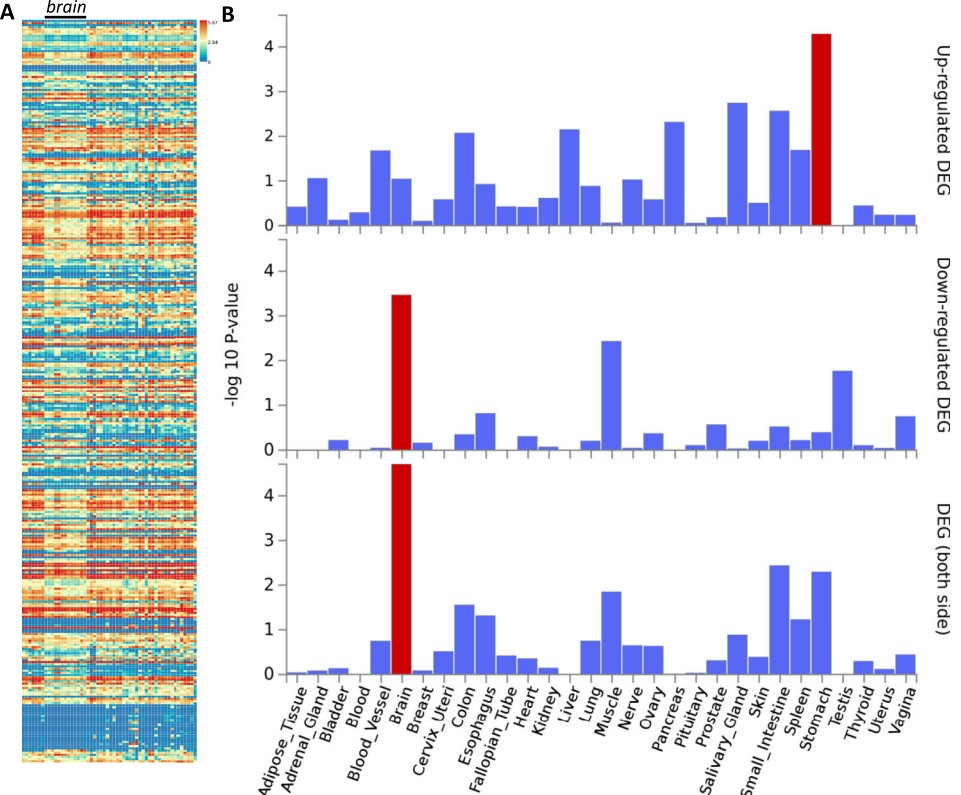

**Fig. 8 FUMA GENE2FUNC analysis of 354 glycosylation enzymes and related genes in humans revealed a specific downregulation in the brain. A** Heat map demonstrating expression pattern of all glycosylation genes in humans, with a black bar above the 13 columns representing brain regions. **B** Tissue specific analysis showing down-regulation in the brain compared to all other 29 tissue types, with significant differentially expressed gene (DEG) sets using a two-sided t-test ($P_{bon} < 0.05$ and absolute fold change $\geq 0.58$) highlighted in red.

evident between cortex and cerebellum, though in general, the majority of brain regions show an overall downregulation of glycosylation genes (Supplementary Fig. 6).

## Discussion

The brain contains millions of cells and billions of connections, creating an unparalleled level of complexity in its development, organization, and regulation. Glycosylation plays a critical role in the establishment and maintenance of this elaborate network, emphasizing the need to understand the unique glycan species involved. Utilizing MALDI-TOF glycomics, MS/MS, lectin blotting, and RNA sequencing, we have generated a comprehensive map of the predominant N- and O-linked protein glycans across multiple brain regions and both sexes of mice. Our findings illustrate a relative simplicity of these structures in the brain and a global downregulation of the pathway, suggesting protein glycan synthesis is tightly controlled.

The overall pattern of brain glycans, in both mouse and human samples, was markedly distinct from those of other tissues. N-glycomics identified predominantly high-mannose and fucosylated/bisected structures in the mouse brain, with few galactosylated, sialylated, or multi-antennary species present, consistent with our results from lectin blotting, as well as a recent study of N-glycans analyzed across brain regions in adult mice and in the prefrontal cortex during development using liquid chromatography MS[43]. We detected both O-GalNAc and O-Man glycans in the brain, though the former were several-fold more abundant across all brain regions. O-GalNAc and O-Man glycans consisted primarily of unbranched core 1 structures (as opposed to extended core 2), and in contrast to N-glycans, were almost entirely

sialylated. RNA sequencing suggests that gene expression is at least in part responsible for the unique glycome profile observed in the brain. The cortex, hippocampus, striatum, and cerebellum have overall similar glycomes; however, we identified several glycans, glycan classes, and glycosylation enzymes that differ significantly between brain regions, emphasizing the need to study these regions independently. The cerebellum was the most unique, with more complex, branched, and hybrid N-glycans, as well as the largest proportion of O-Man species. We detected relatively few differences in brain protein glycosylation between sexes, in contrast to their distinct plasma N-glycomes, suggesting more conserved regulation of glycosylation in the brain compared to other tissues, though additional female brain O-glycan samples will be informative in confirming sex-specific differences.

The relative simplicity of brain N-glycans is surprising considering their essential physiological roles. High-mannose N-glycans are often considered immature precursor structures but comprise the majority of all N-glycans in the brain. These structures appear to be mature, as they have been detected on the plasma membrane of neurons, as well as on extracellular matrix proteins[77–80]. Some studies have demonstrated that these glycans are involved in cell-cell recognition and homeostatic maintenance, governing the interaction properties of NCAM and basigin and influencing neurite and astrocytic outgrowth[77,81,82]. High-mannose N-glycans are also recognized by the mannose receptor (CD206), a microglia specific receptor that can regulate endocytosis and thus may play a role in synaptic pruning[83–86].

Of the ~30% of N-glycans in the brain which are not high-mannose structures, the majority (80–90%) are bisected. This may contribute to the lack of extended glycans in the brain, as

bisection has been shown to impede subsequent modifications of N-glycans, including galactose and sialic acid, since the additional GlcNAc residue may alter the glycan conformation to prevent interactions with glycosyltransferases[87,88]. *Mgat3* knockout mice develop normally while lacking bisected structures and show a greater relative abundance of complex and modified N-glycans[35]. However, high-mannose structures still comprise the majority of N-glycans in the brain of *Mgat3*$^{-/-}$ mice, suggesting this molecular brake is only one mechanism in place leading to a low abundance of complex N-glycans.

O-GalNAc glycans can be extensively modified in other organs[89,90] but are limited to mostly sialylated core 1 structures in the brain. Though they comprise the majority of brain O-glycans, the functional roles of O-GalNAc structures are not well understood in the nervous system. A recent case series identified mutations in GALNT2, one of the 20 enzymes capable of attaching the core GalNAc residue to a serine or threonine, as the cause of a novel CDG[91]. Symptoms include intellectual disability, epilepsy, insomnia, and brain MRI abnormalities, and rodent models of *Galnt2* knockout also displayed neurologic abnormalities consistent with a functional role of Galnt2-mediated glycosylation in the brain.

O-Man structures are better understood in terms of their protein carriers and physiological functions, despite their lower abundance[37,38,92]. One common carrier is α-dystroglycan, studied extensively in congenital muscular dystrophies, though knockout studies have shown that there are many other proteins modified by O-Man in the brain[37,93]. Extended O-Man glycans, including those harboring the HNK-1 and Le$^X$ epitopes, have been identified on components of perineuronal nets, extracellular matrix structures involved in cell adhesion and neurite outgrowth[94–96]. Core M2 glycans have only been reported in the brain, where the key synthetic enzyme MGAT5B is highly enriched, and regulate remyelination, astrocyte activation, and oligodendrocyte differentiation[97–101]. A unique mono-O-mannose glycan on members of the cadherin family has been recently described, and is necessary for the cell-adhesion function of these proteins[102,103]. This O-Man attachment is catalyzed by a novel family of O-mannosyltransferases known as TMTC1-4, rather than the canonical POMT-initiated O-mannose pathway, and is not extended further than the core Man residue[104,105]. We did not identify mono-O-man or other monosaccharide modifications such as mono-O-Fuc or mono-O-GlcNAc, despite brain expression of their synthetic enzymes (*Tmtc1-4*, *Pofut1-2*, and *Ogt*). Such modifications may be present at a lower abundance relative to extended O-GalNAc and O-Man glycans in the brain, as previous studies have primarily used enrichment strategies for their isolation[104,106,107].

While less than 3% of brain N-glycans are modified by sialic acid, almost all of the O-glycans detected in this study are sialylated. *St3Gal2* and *St6galnac6* are among the highest expressed sialyltransferases in the brain and involved in the synthesis of the abundant disialylated core 1 O-GalNAc structure (*m/z*: 1257), which may account for the imbalance in O-glycan vs N-glycan sialylation. Despite its decreased relative abundance on brain glycoproteins[56], sialic acid has been studied extensively in the context of brain development and disease[33]. Sialic acid is a regulator of phagocytosis, as microglia express several siglec-type receptors that recognize sialic acid and trigger an inhibitory response in the cell upon binding[108,109]. Enzymatic removal of sialic acid from neurons in culture decreases siglec binding, increases engulfment by microglia, and potentiates complement deposition, a key regulatory step in microglial-mediated synaptic pruning[110–114]. Another carrier of sialic acid in the brain is PSA-NCAM, which can harbor up to 400 sialic acid residues and is critical in brain development and neuronal migration[23,115]. We did not identify this structure in our samples likely due to its large size and low abundance in the adult brain[116–118].

In sum, we present a comprehensive picture of protein N- and O-glycosylation in the mouse brain. Our results highlight unique glycan compositions and distinct regulatory mechanisms across several brain regions, tissue types, and sexes in one of the largest sample sizes to date. We highlight the value of complementary analyses as several prior assumptions on the identity, composition, and linkage of glycans in the brain were incorrect when relying on a single method. For example, we defined several N-glycans as bisected and hybrid (*m/z*: 1836, 2244) that were previously described with different antennarity and galactosylation[119], or as LacdiNAc structures[43]. Further, we correlated the observed glycan structures with the presence (*Mgat3* for bisection) or absence (*Ggta1* for α-Gal) of their synthetic enzymes. Given the limitations unique to each method, such as the semi-quantitative nature of MALDI-MS and the dynamic range of western blotting, comparison between analytic techniques should be interpreted with caution, particularly for the study of low abundance molecules. Additional quantitative measures of glycan concentration will strengthen the findings of a single analytical approach such as MALDI-MS glycomics. We have applied such techniques to the study of brain glycosylation changes caused by a single point mutation associated with schizophrenia using fluorescent glycan derivatization[56]. These data provide additional supportive evidence of the conclusions drawn in this study, including observed differences in the relative abundance and sialylation between N- and O- glycans. Future studies addressing qualitative and quantitative measures of glycosylation should employ several independent yet complementary analytical methods in order to draw meaningful conclusions.

Here we emphasized the most abundant N- and O-glycans in the brain and their potential physiological roles, but this makes no assumption of the function or importance of structures that exist at very low abundance. Subtle changes in glycosylation can lead to major consequences at the protein, cell, and circuit level, so it is essential to understand how such variation is regulated at the genetic[20], epigenetic[120], transcriptional[121], developmental[41,50], regional[40,52,122], and organismal levels[67,68,123]. The contribution of glycosylation to health and disease has been appreciated in many contexts, especially the nervous system[124]. In addition to neurologic symptoms of CDGs[16], complex neuropsychiatric phenotypes are linked to glycosylation[19,20,125]. For example, several glycosyltransferases and a missense variant in *SLC39A8* are associated with schizophrenia, emphasizing the need for a more detailed understanding of protein glycosylation as it relates to development and disease in the brain[17].

## Methods

**Samples**. Fresh (unperfused) postmortem mouse brain samples were harvested from wild-type mice on a C57BL/6J background originally from The Jackson Laboratory (Cat#000664) after euthanasia with $CO_2$, as well as a sample of whole blood for plasma analysis. Mice from both sexes were used in this study and were 12 weeks old at the time of tissue harvest, sample size specified for each experiment. All mice were housed and maintained in accordance with the guidelines established by the Animal Care and Use Committee at Massachusetts General Hospital under protocol #2003N000158. Human Brain Cerebral Cortex Whole Tissue Lysate was purchased from Novus Biologicals (#NB820-59182), with 1mg used for glycomic analysis as described below.

**Brain dissection**. Following euthanasia with $CO_2$, the whole mouse brain was removed and placed on a clean ice-cold plastic surface and rinsed with PBS at 4 °C. Four brain regions (frontal cortex, hippocampus, striatum, cerebellum) were isolated from each hemisphere using blunt dissection and placed in 1.5 mL conical tubes, snap frozen in liquid $N_2$, and stored at −80 °C until further use.

**Tissue lysis**. Frozen brain tissue was lysed in 500 μL ice-cold lysis buffer (50 mM TRIS, 150 mM NaCl, 1.0% w/v Triton-X-100, pH 7.6) with protease inhibitor (Roche #46931320019) and dissociated using a hand-held motorized pestle (Kimble #749540), followed by 2 brief pulses of sonication for 10 seconds with a microtip (Qsonica Q700). An additional 500 μL of lysis buffer was added to bring

the volume to 1 mL, and protein concentration was analyzed using the Pierce BCA Protein Assay Kit (ThermoFisher Scientific #23255).

**Isolation and purification of glycoproteins**. Brain glycoproteins were purified according to standard protocols readily available through the National Center for Functional Glycomics website (https://ncfg.hms.harvard.edu/protocols). All buffers were made fresh daily. In brief, 2 mg of protein lysate per sample was dialyzed in 3.5 L of 50 mM ammonium bicarbonate 3 times at 4 °C over 24 h using snakeskin dialysis tubing with a molecular cut-off between 1 and 5 kDa (ThermoFisher Scientific #68035). Samples were lyophilized and then resuspended in 1 mL of 2 mg/mL 1,4-dithiothreitol (DTT) dissolved in 0.6 M TRIS (pH 8.5) and incubated at 50 °C for 1.5 h, followed by addition of 1 mL of 12 mg/mL iodoacetamide in 0.6 M Tris (pH 8.5) and incubated at room temperature for 90 min in the dark. Samples were again dialyzed as described above, lyophilized, and resuspended in 1 mL of 500 µg/ml TPCK-treated trypsin in 50 mM ammonium bicarbonate and incubated overnight (12–16 h) at 37 °C. Trypsin digestion was stopped by the addition of ~2 drops 5% acetic acid, and samples were added to a C18 Sep-Pak (200 mg) column (Waters, #WAT054945) preconditioned with one column volume each of methanol, 5% acetic acid, 1-propanol, and 5% acetic acid. Reaction tubes were washed with 1 mL 5% acetic acid and added to the column, followed by an additional 5 mL wash of 5% acetic acid. Each column was placed in a 15 mL glass tube, and glycopeptides were eluted using 2 mL of 20% 1-propanol, 2 mL of 40% 1-propanol, and 2 mL of 100% 1-propanol. The eluted fractions were pooled and placed in a speed vacuum to remove the excess organic solvent, followed by lyophilization.

**Release and purification of protein N-glycans**. Lyophilized glycopeptides were resuspended in 200 µL of 50 mM ammonium bicarbonate and incubated with 3 µL of either PNGase F (New England Biolabs, #P0704) or Endo H (New England Biolabs, #P0702S) at 37 °C for 4 h, then overnight (12–16 h) with an additional 5 µL of the enzyme at 37 °C. C18 Sep-Pak columns (200 mg) were preconditioned with one column volume of methanol, 5% acetic acid, 1-propanol, and 5% acetic acid and placed in 15 mL glass tubes. Digested samples were loaded onto preconditioned columns, collecting all flow-through, and N-glycans were eluted with 6 mL of 5% acetic acid. Eluted fractions were pooled and placed in a speed vacuum to remove the organic solvents and lyophilized overnight. Glycopeptides remaining on the C18 columns were eluted using 2 mL of 20% 1-propanol, 2 mL of 40% 1-propanol, and 2 mL of 100% 1-propanol, placed in a speed vacuum to remove the organic solvents and lyophilized for O-glycan processing.

**β-elimination and purification of O-glycans**. After removing N-glycans from glycopeptides, O-linked glycans were removed using a β-elimination reaction according to the standard protocols available through the National Center for Functional Glycomics (https://ncfg.hms.harvard.edu/protocols). In brief, lyophilized N-glycan-free glycopeptides were resuspended in 400 µL of 55 mg/mL NaBH4 in 0.1 M NaOH solution and incubated overnight (12–16 h) at 45 °C. β-elimination reaction was terminated by adding 4–6 drops of glacial acetic acid to each sample. Desalting columns were prepared using Dowex 50W X8 ion exchange resin with the mesh size of 200–400 (Sigma-Aldrich, #44519) in small glass Pasteur pipettes and washed with 10 mL of 5% acetic acid. Acetic acid-neutralized samples were loaded onto columns, collecting flow through in 15 mL glass tubes. Columns were washed with an additional 3 mL of 5% acetic acid and all flow-through was pooled, placed in a speed vacuum to remove the organic solvent and lyophilized. Dried samples were resuspended in 1 mL of 1:9 acetic acid:methanol solution (v/v = 10%) and dried under a stream of nitrogen, repeating this step an additional three times. C18 Sep-Pak columns (200 mg) were preconditioned with one column volume of methanol, 5% acetic acid, 1-propanol, and 5% acetic acid and placed in 15 mL glass tubes. The dried samples were resuspended in 200 µL of 50% methanol and loaded onto the conditioned C18 columns, collecting flow-through. Columns were washed with 4 mL of 5% acetic acid and all flow-through pooled, placed in a speed vacuum to remove the organic solvents and lyophilized.

**Preparation and isolation of plasma N-glycans**. Blood samples were collected following $CO_2$ euthanasia and decapitation in a microtainer tube (BD, #365967), and plasma was separated by centrifugation and stored at −80 °C until use. Plasma N-glycan profiling was performed as described previously[73]. In brief, 5 µL of mouse plasma was lyophilized, resuspended in 20 µL 1X Rapid PNGase F buffer (NEB #P0710S), and denatured at 70 °C for 15 min After cooling to room temperature, 1 µL of Rapid PNGase F was added, and incubated at 50 °C for 60 min C18 Sep-Pak columns (50 mg, Waters, #WAT054955) were preconditioned with one column volume of methanol, 5% acetic acid, 1-propanol, and 5% acetic acid and placed in 1.5 mL tubes. PNGase F-treated samples were resuspended in 100 µL of 5% acetic acid and added to the preconditioned columns, collecting all flow-through. The reaction tube was washed with an additional 100 µL of 5% acetic acid which was added to the column, followed by 1 mL of 5% acetic acid, and the entire flow-through was placed in a speed vacuum to remove the organic solvents and lyophilized prior to permethylation as described below.

**Permethylation of N- and O-glycans**. Using a clean, dry mortar and pestle, 21 pellets of NaOH were ground and dissolved into 12 glass pipettes volumes (~3 ml) of DMSO. A fresh slurry of NaOH/DMSO was made daily. One mL of the slurry was added to the lyophilized N- and O-glycans in addition to 500 µL of iodo-methane (Sigma Aldrich, #289566). Samples were tightly capped and placed on a vortex shaker for 30 min at room temperature. After the mixture became white, semi-solid, and chalky, 1 mL ddH2O was added to stop the reaction and dissolve the sample. 1 mL of chloroform and an additional 3 mL ddH2O were added for chloroform extraction and vortexed followed by brief centrifugation. The aqueous phase was discarded, and the chloroform fraction was washed three additional times with 3 mL ddH2O. Chloroform was then evaporated in a speed vacuum. Permethylated glycans were resuspended in 200 µL of 50% methanol and added to a C18 Sep-Pak (200 mg) column preconditioned with one column volume each of methanol, ddH2O, acetonitrile, and ddH2O. The reaction tubes were washed with 1 mL 15% acetonitrile and added to the column, followed by an additional 2 mL wash of 15% acetonitrile. Columns were placed into 15 mL glass round-top tubes, and permethylated glycans were eluted with 3 mL 50% acetonitrile. The eluted fraction was placed in a speed vacuum to remove the acetonitrile and lyophilized overnight.

**MALDI-TOF-MS**. Permethylated glycans were resuspended in 25 µL of 75% methanol and spotted in a 1:1 ratio with DHB matrix on an MTP 384 polished steel target plate (Bruker Daltonics #8280781) as previously described[73]. MALDI-TOF MS data was acquired from a Bruker Ultraflex II instrument using FlexControl Software in the reflective positive mode. For N-glycans, a mass/charge ($m/z$) range of 1000–5000 kD was collected, and for O-glycans, a range of 500–3000 kD. Twenty independent captures (representing 1000 shots each) were obtained from each sample and averaged to create the final combined spectra file. Data was exported in .msd format using FlexAnalysis Software for subsequent annotation. Tandem MS (MS/MS) data were collected using the same instrument for both N- and O-glycans, using the LIFT positive mode, and a $+/- 1$ Da range from the predicted parent $m/z$, and again represent the sum of twenty independent captures.

**N- and O-glycan analysis**. Glycans of known structure corresponding to the correct isotopic mass which had a signal to noise ratio greater than 6 (S/N) in at least one brain region averaged over the grouped samples were annotated using mMass software[126]. This resulted in 95 brain N-glycans, 26 brain O-glycans, and 29 plasma N-glycans. The relative abundance of each glycan was calculated as the signal intensity for each isotopic peak divided by the summed signal intensity for all measured glycans within a spectrum. Although using the isotopic mass for quantification may underestimate the relative abundance of larger glycans given the increased incorporation of Carbon-13, the majority of N- and all of O- glycans in the brain are best represented by the isotopic peak ($m/z < 2040$). Structural assignment of glycans was based on MS/MS results, enzyme sensitivity (PNGase F, Endo H), previously confirmed structures[35,37,58], and deductive reasoning when able. MS/MS data was annotated by comparing resultant $m/z$ peaks to the predicted values for fragment ions with up to three bond breaks from all possible parent structures using GlycoWorkbench[127]. Brain protein glycans were grouped into different categories based on shared components, such as monosaccharide composition, antennary, etc., and the summed abundance of each category was compared across brain regions and sexes. All glycan structures are presented according to the Symbol Nomenclature for Glycans (SNFG) guidelines[128,129] and were drawn using the GlycoGlyph online application[130].

**Lectin blotting**. Brain lysate from the cortex and cerebellum of male mice, were precleared using magnetic streptavidin beads (New England Biolabs, #S1420S) at a 1:2 ratio of µg protein to µL beads to decrease background binding resulting from high levels of biotin-bound carboxylases in the brain. After 1-h incubation at room temperature, beads and biotin-bound proteins were precipitated using a magnetic tube rack, and the supernatant was removed for further analysis. PNGase F sensitivity was determined by incubation of 100 µg protein with 5 µL PNGase F (New England Biolabs, #P0704S) at 37 °C for 1 h. Lysates were prepared with 4X Sample Loading Buffer (Li-COR, 928–40004) with 10% v/v β-mercaptoethanol, and denatured for 10 min at 95 °C. For each gel, 15 µg protein was loaded per well (NuPAGE 4 to 12% Bis-Tris, 1.0 mm, Mini Protein Gel, 12-well, ThermoFisher, NP0322). In addition to 2 µL Chameleon Duo Pre-Stained Protein Ladder (LiCOR, 928–60000), 50 µg of human plasma was loaded as a positive control; plasma is ~60% is non-glycosylated albumin, thus ~20 µg plasma glycoprotein per lane. Gels were run using the MiniProtean Tetra Electrophoresis System (BioRAD, 1658004) at 140 mV for 1 h. Proteins were transferred to nitrocellulose membranes (ThermoFisher, IB23003) using the iBlot Dry Blotting System (ThermoFisher, IB1001). Membranes were then incubated in 5% BSA in TBS-Tween 0.1% for 1 h, followed by incubation with biotinylated lectins (Vector Labs: AAL B-1395, SNA B-1305, GNL B-1245, PHA-E B-1125, RCA B-1085, ConA B-1105) at a 1:1,000 dilution (1:20,000 for ConA) and 1:2,000 dilution of mouse antiactin antibody (Abcam, ab8226) in 5% BSA in TBS-Tween 0.1%, overnight at 4 °C on a rocking platform shaker. Membranes were washed three times in TBS-Tween 0.1% for 5 min, and then incubated with fluorescent conjugated streptavidin IRDye 800CW (LiCOR, 926–32230) and Goat anti-Mouse IgG IRDye 680RD

(LiCOR, 925–68070) at 1:25,000 dilution in 5% BSA in TBS-Tween 0.1% for 30 min protected from light. Membranes were again washed three times in TBS-Tween 0.1% for 5 min and imaged using a LiCOR Odyssey CLx Imaging System and analyzed using LiCOR Image Studio Software. An identical unprobed membrane was incubated with Revert 700 Total Protein Stain (LiCOR, 926–11011) according to manufacturer's protocol. A comprehensive characterization of biotinylated lectin binding specificity by glycan microarray can be found on the National Center for Functional Glycomics website (https://ncfg.hms.harvard.edu/ncfg-data/microarray-data/lectin-quality-assurancequality-control).

**RNA sequencing.** Using the contralateral hemisphere of 4 male mouse brains used in glycomics and lectin blotting experiments, RNA from snap-frozen cortex and cerebellum was purified using the RNeasy Lipid Tissue Mini Kit (QIAGEN, 74804) per manufacturer's protocol. RNA-seq libraries were prepared from total RNA using polyA selection followed by the NEBNext Ultra II Directional RNA Library Prep Kit protocol (New England Biolabs, E7760S). Sequencing was performed on Illumina HiSeq 2500 instrument resulting in approximately 30 million of 50 bp reads per sample. Sequencing reads were mapped in a splice-aware fashion to the mouse reference transcriptome (mm9 assembly) using STAR[63]. Read counts over transcripts were calculated using HTSeq based on the Ensembl annotation for GRCm37/mm9 assembly and presented as Transcripts Per Million (TPM)[62]. A subset of 269 known glycosyltransferases, glycosylhydrolases, sulfotransferases, and glycan-related genes was created, and differences in expression level between cortex and cerebellum were performed as described below.

**Human RNA comparison and FUMA analysis.** Performed utilizing publicly available gene expression data from the Genotype-Tissue Expression (GTEx) Portal, Version 8 (https://gtexportal.org). 354 known glycosyltransferases, glycosylhydrolases, sulfotransferases, and glycan-related genes IDs from humans were used as input into the GENE2FUNC platform of FUMA, which utilizes the GTEx v8 data of both 30 general tissue types, with all brain regions summarized as one tissue type, and 54 specific tissue types that include 13 individual brain regions. Data is presented alphabetically, with differentially expressed gene sets shown in red after Bonferroni correction with corrected $p < 0.05$ and absolute fold change $\geq 0.58$ using the standard two-sided t-test described on the GENE2FUNC platform.

**Statistical analysis.** For glycomic analyses, statistical analysis of individual and groups of glycans was performed with Microsoft Excel Version 16.35. The abundance of individual glycans and glycan classes were compared between brain regions using single factor ANOVAs. Unpaired two-tailed $t$ tests assuming unequal variance were performed for sex comparisons of individual N-glycans and glycan classes from the cortex, cerebellum. Significance thresholds for ANOVAs and $t$ tests were applied at $p < 0.05$ (*), $p < 0.01$ (**), and $p < 0.001$ (***). A simple regression was performed between O-glycans modified with NeuAc or Fuc using GraphPad Prism v8.4.2. The EdgeR method was used for differential expression analysis of RNAseq data with gene cutoffs of 2-fold change in expression value and false discovery rates (FDR) below 0.05 as previously described using EdgeR and Python software[64]. Global glycosylation gene regulation in humans was analyzed using the FUMA GWAS GENE2FUNC online tool, which identified significantly up- or downregulated differentially expressed gene sets across human tissue types with a Bonferroni corrected $p$ value $< 0.05$.

**Reporting Summary.** Further information on research design is available in the Nature Research Reporting Summary linked to this article.

## Data availability

The data generated in this study are included in this published article and its supplementary information files. Source data are provided with this paper. The raw MS glycomics data generated in this study have been deposited in the GlycoPOST[131] database under accession code GPST000213 (wild-type and A391T mutant glycomics data[56]). The RNAseq data generated in this study have been deposited in the NCBI's Gene Expression Omnibus[132,133] under GEO Series accession number GSE184516 (wild-type and A391T mutant RNAseq data[56]). Human gene expression data is publicly available from the Genotype-Tissue Expression (GTEx) Portal, Version 8 (https://gtexportal.org). Protocols for glycomics analysis are publicly available through the National Center for Functional Glycomics (https://ncfg.hms.harvard.edu/protocols). Source data are provided with this paper.

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

## Acknowledgements

This work was supported by a foundation grant from the Stanley Center for Psychiatric Research at the Broad Institute of Harvard/MIT (awarded to RGM) and NIH grants P30DK040561 (awarded to R.I.S) and P41GM103694 (awarded to RDC). RGM is supported by T32MH112485.

## Author contributions

SEW performed glycomics experiments, statistical analysis, assisted in RNA analysis, and wrote the manuscript with RGM. MN performed lectin blotting, assisted in RNA analysis, and wrote portions of the manuscript. SL assisted with experimental methods and advised on analysis of glycans. MC performed analysis of RNAseq data. RX provided mouse samples for analysis and assisted with project design. RS oversaw RNAseq analysis and wrote related portions of the manuscript. EMS initiated the project and coordinated collaborations. JWS co-supervised SEW and helped conceptualize the project. RDC co-supervised SEW and MN, oversaw all experimental analyses, and helped conceptualize the project. RGM co-supervised SEW and MN, performed glycomics experiments, lectin blotting, RNA purification and analysis, statistical analysis, helped conceptualize the project, and wrote the manuscript. All authors contributed feedback and edits to the manuscript.

## Competing interests

R.J.X. is a cofounder and equity holder of Celsius Therapeutics and Jnana Therapeutics and consultant to Novartis. These companies did not provide support for this work. J.W.S. is a member of the Scientific Advisory Board of Sensorium Therapeutics and has received honoraria for an internal seminar at Biogen, Inc and Tempus Labs. These companies did not provide support for this work. The remaining authors declare no competing interests.

**Additional information**

