## [Peer Review File · Nature Communications]

REVIEWER COMMENTS

Reviewer #1 (Remarks to the Author):

In this manuscript, the authors studied the glycosylation from mouse brain samples. N- and O-glycosylation were characterized between the mouse brain region and gender through MALDI-TOF MS, and lectin blotting. In addition, through RNA sequencing, the expression patterns of known glycan-related genes based on the Carbohydrate Active Enzymes database (CAZy) were investigated. This is precisely the type of research the needs to be done to understand the biochemistry of the brain and the results are very interesting.

However, the question arises as to whether the number of samples is sufficient to produce meaningful results. In particular, it was considered that no significant data analysis was performed with a very small number of samples (male:3, female: 2) in O-glycan analysis between sexes. Even the numbers for human samples are not mentioned anywhere in the manuscript. In addition, the link between the findings begins with brain glycan analysis, so accurate qualitative and quantitative analysis of glycans is important. However, since the authors analyzed glycans with MALDI-TOF MS, they did not take into account the various isomeric structures of glycans, an important characteristic. As an example, the potential physiological roles of abundant N-glycans and O-glycans in the brain obtained via MALDI were highlighted, whereas structures present in very low content were not considered. As is well known, since there is a limit to obtaining quantitative data through MALDI-MS analysis, additional discussions on quantitative experiments and results are necessary. They should state early on and in the results how many mice were used in the study.

The authors mentioned in the introduction that so far no brain region specific analysis has been performed. However, there was a reference recently published that comprehensively covers brain glycosylation (Spatial and temporal diversity of glycome expression in mammalian brain, PNAS USA 117, 28743–28753 (2020)). Glycans were profiled from nine different brain regions in mouse and glycome diversity were compared. This should be properly described in the manuscript.

Reviewer #2 (Remarks to the Author):

This work presents comprehensive glycomic analysis of 4 mouse brain regions and couples this analysis to orthogonal validation by lectin blotting and to comprehensive transcriptomic

characterization of genes involved in glycosylation. The combined datasets are important for the field because they are performed in a robust and standardized fashion and are done all on the same sample sets. Other works in the field have done one of these analyses or another, but here they are all brought together in a well-structured and clearly written work. The manuscript reports 3 major findings. First, N-glycans in brain are under-processed and lack significant regional diversity. Second, O-glycans in brain are also limited in structural complexity and lack significant regional diversity. Transcripts for glycozymes that encode for glycan biosynthetic enzymes are consistent with glycomic profiles and, in general, exhibit less potential for generating structural diversity. Previous work has suggested that this may be the case, especially in relation to the abundance of high-mannose and pauci-mannose glycans, but this work goes further, tying together analytic data in support of the hypothesis that brain glycosylation is tightly regulated and restricted in diversity. The data provides an important platform for future analysis of mouse models of human brain disorders and provides a roadmap for evaluating human brain glycomic data. Attention to the following specific comments would strengthen the clarity, interpretability, and presentation of the data.

1. Technical detail. Were the brains perfused to remove blood before dissection? If not, the authors should comment on the extent to which the glycomic profiles are influenced by blood elements.

2. It is assumed by this reviewer, but not stated anywhere, that the masses presented in the various tables, figures, and supplemental material are of the permethylated, perhaps sodiated, forms of the glycans. Please provide this information early in the manuscript.

3. Fig. 2 and others. Where more than one possible glycan structure is indicated for a particular mass (usually bracketed together), it is not always clear whether both structural isomers were detected in MS/MS or both structural isomers are possible but the data does not distinguish between the two. Are the two possibilities indicated to be considered “this and this,” vs. “this or this?”

4. Fig 2, panel C. What does “mono-antennary” mean? Is this different from hybrid? If so, how? Are there more than one “mono-antennary” glycans in the glycan set used for the heat-map? Pauci-mannose with extension on the 3-arm?

5. Fig. 3, panel B. The justification for using the “#” placeholder is not clear. What happened to the signal intensity associated with the F2A1G1BH4 glycan? Also, for this entire figure, it is not clear what the values on the y-axis (x105 au) are meant to represent. This experiment is, I believe, a MALDI analysis of permethylated glycans released by the indicated digestions. It would be useful to provide some text that helps the reader understand how to compare the disparities in signal strengths across the three panels.

6. Fig. 5, panel D. The figure legend describing panel D is missing some words or a phrase.

7. Fig. 7, panel E. The Fut8 reaction is over-simplified for presentation. It is not likely that Fut8 adds Fuc to the paucimannose glycan directly. Rather, the Fuc was probably added to a precursor that has been degraded to the paucimannose form. Would suggest that the MGAT1/GnT1 product be used for this illustration.

8. The top 10 N-glycans (of 94) were presented in Fig. 2 and the top 15 in Fig. 3. Sorting the glycans for abundance in Table S1, it looks like the m/z values at 1345 and 2838, which were excluded from the top 10, are more highly abundant than at $m/z=2244$, which was included. Further clarification of why certain glycans were included for some analysis and excluded for others would be useful. The authors mention that all of the glycans (assuming all of those in Table S1) were detected and quantifiable with a signal to noise of at least 6 to 1. Did the authors aggregate the features (number of antennae, core fucosylation, complex, hybrid, etc.) across all of the glycans, not just the top 10 or 15? If so, does a broader aggregation identify more subtle regional diversity? Same for O-linked glycans?

9. In the discussion, p. 20 beginning with line 446, the authors indicate that this study adds value on top of previous studies because it clarifies that “several prior assumptions on the identity, composition, and linkage of glycans in the brain were incorrect....” However, no reference to these previously incorrect assignments are provided, neither are specific examples cited. This claim should be elaborated upon or removed.

10. The authors mention that data will be made available upon “reasonable request.” Have the authors considered submitting their raw MS and MS/MS data to an established repository, such as GlycoPost? Whether it is required by this journal or not, deposition to such a repository would be a very valuable contribution to the field. If this dataset, as the authors propose, will provide essential baseline data for future studies, it seems even more important for the raw data to be broadly accessible for future mining and interpretation. Likewise, it would be equally useful for the glycan compositions described in Table S1 (and in other places in the manuscript) to be associated with appropriate database accession numbers (GlyTouCan IDs). This association will greatly facilitate the uptake and dissemination of this data by various international glycan databases and knowledgebases. Again, I am unaware of whether this is required by the journal, but it would be a significant service to the field and a valuable addition to the authors’ dataset.

Reviewer #3 (Remarks to the Author):

This is an excellent manuscript analysing the N- and O-protein glycosylation in the brain including looking at different sections and sex differences. The data indicates a unique glycosylation of the brain compared to plasma (not really surprising!) and its synthetic implications and the relative uniformity will be very interesting to explore further. The mass spectrometric analysis is done in parallel by analysing the transcriptome and lectin staining data in a mouse model. Public genome databases are then analysed for the distribution of gene expression of the over 350 glycosylation enzymes in human brain relative to other organs.

There are some questions which could be elaborated upon:

- Fig 2 c– the p- values? In the Anova section only one star is shown in the figure whereas up to 3 stars are seen in Figure legend. Heat map seems to indicate that at least paucimannose, tetra antennary , sialylation should be significantly different between cortex and cerebellum?

- Fig 3 nomenclature used should be defined e.g FA1B etc

- What is the definition of the paucimannose structures seen?

Essentials gives as M3-4 GlcNac2, CHEBI gives M1-4Fuc0-1GlcNac2

M5-9 would be considered oligomannose. The only putative pauci shown FM3 is not visible in the spectra and the other smaller varieties would also not be seen.

- Low sialic acid on N-glycans – was there a control for detection of sialylation after permethylation/MALDI?

- “Simple linear regression of fucosylated vs sialylated O-glycans showed a strong and highly significant negative correlation in both O-GalNAc and O-mannose glycans, suggesting competition between these modifications in the brain” - Inverse correlation does not necessarily be explained by competition?

- Not clear if figs 2 and 3 and 4 are mouse or human brain derived – it is indicated neither in text or figure legend ? This needs to be made clearer throughout first sections? And were they the same sex?

- O-glycans were highly sialylated – N glycans were not – were there specific sialyltransferase transcriptome expression differences?

- How is presence of NeuGC explained on the O-glycans when you state for the NeuGc absence on the N-glycans “and the lack of expression of the enzyme which converts NeuAc to NeuGc in the brain”?

- Line 215 – this is not shown in Fig 3C? Do you mean Fig 4C?
- Fig 5 – why are there different n= for the N-linked and O-linked analyses?
- Fig 6: SNA binding is interesting as seems to indicated O-linked sialylation on a wide range of proteins?
- Would the MS method see mono– O-monosaccharides?
- Have you considered polysialylation of the O-glycans? What is the "low abundance (of PSA) in the adult brain" in quantitative terms?
- "We highlight the value of complementary analyses as several prior assumptions on the identity, composition, and linkage of glycans in the brain were incorrect when relying on a single method" – would be good to give an example here.

Methods:

- What Triton X was used for lysis?
- Where were human brain sections sourced?

Reviewer #1 (Remarks to the Author):

However, the question arises as to whether the number of samples is sufficient to produce meaningful results. In particular, it was considered that no significant data analysis was performed with a very small number of samples (male:3, female: 2) in O-glycan analysis between sexes.

Our group sample sizes are based on a power calculation from our MALDI method detect limit to measures differences in plasma glycosylation in carriers of a common genetic variant (Mealer, *et al.*, Sci. Rep 2020). Based on this data, and assuming an effect size of ~10% on glycosylation phenotypes in a mouse model, we determined that groups of 4-5 would be sufficient to detect differences using MALDI-MS as described in the complementary manuscript which is also under review. We applied this group size to each of the brain regions described for both sexes for glycomics, resulting in a total number of wild-type brain samples measured by MALDI-MS of ~50 for N-glycans. Our lab has performed MALDI-MS glycomic analysis of nearly 50 other mouse samples containing a single nucleotide polymorphism associated with schizophrenia from both sexes and the same brain regions and found striking overall similarity (described in complementary manuscript), consistent with the data and conclusions presented in this manuscript. In addition, we have analyzed dozens of human samples using both MALDI-MS and LC-MS in our lab for independent projects not described here, which have all produced comparable results. Based on this experience, we are confident in the ability of our methods to produce meaningful results with this sample size, and our study differs from many prior brain glycomics experiments where a single brain was used for the entirety of the results presented without consideration for sex, region, or age (for example, see Riley *et al.*, 2019, Nat. Comm.), or a whole brain from a small number of only male mice was used for the entirety of the study (see Shen *et al.*, 2021, Nat. Methods).

The reviewer highlights one group with the smallest sample size, namely the O-glycans of cortex in females where $n = 2$. We initiated the experiment with groups of at least 6 mice per sex. During the purification, specifically that of O-glycans, which are the least abundant in the brain, the ~1-month protocol including dialysis of lipids, trypsinization, removal of N-glycans with PNGase F, B-elimination of O-glycans, purification, permethylation, and isolation for MS analysis, we have observed that some samples do not produce any detectable signal for MS. Presumably the product is lost somewhere during the isolation due to a technical issue. We aim to have at least 3 samples/group to perform a meaningful comparison (with a mean +/-SEM). Only in this specific group (female wild-type cortex) did we opt to report the data while not perform any statistical comparison or draw preliminary conclusions, but hope including the data for qualitative comparisons will be informative to the reader. In addition, in our complementary paper, the sample size for each region and sex are often doubled if including mutant mice with the common A391T variant. For example, O-glycan analysis of cortex in female mice includes 6 total samples (2 wild type, 4 mutant mice) and show a nearly identical pattern with no significant differences between genotypes. We have considered adding additional new wild-type samples to increase the sample size, but such results would not be directly comparable in isolation to data from other brain regions or RNAseq, and feel the results reported here, all originating from the same wild type mice across multiple platforms, is a strength and core feature of our paper.

Even the numbers for human samples are not mentioned anywhere in the manuscript.

Data from human cerebral cortex was produced from a single commercially available sample from Novus Biological. We have updated the text in results section to make this clearer (Line 96-97), and the methods section (Line 655). The legend for Fig. 1 describes that this is a representative sample and is shown for qualitative purposes.

In addition, the link between the findings begins with brain glycan analysis, so accurate qualitative and quantitative analysis of glycans is important. However, since the authors analyzed glycans with MALDI-TOF MS, they did not take into account the various isomeric structures of glycans, an important characteristic. As an example, the potential physiological roles of abundant N-glycans and O-glycans in the brain obtained via MALDI were highlighted, whereas structures present in very low content were not considered. As is well known, since there is a limit to obtaining quantitative data through MALDI-MS analysis, additional discussions on quantitative experiments and results are necessary.

A major motivation for performing our study is precisely the reviewer's point that accurate comparison across analytic techniques of glycans is essential for both qualitative and quantitative measures, and in particular the isomeric structures and those of low abundance.

Isomeric structures were considered in detail in our study. The reviewer correctly states that single MALDI-MS is unable to distinguish isomeric structures, providing information only on the carbohydrate composition of each glycan. As such, we performed detailed tandem MS (MS/MS) to identify specific isomeric structures as reported in Fig. S1, S2, and S3. In addition, we performed single MS using glycosidase enzymes with different cleavage specificities supporting our MS and MS/MS data (Fig. 3) and blotting with specific carbohydrate-binding lectins for qualitative confirmation (Fig. 6). Tandem MS can only be performed on masses with a reasonable abundance (in our experience > 2% within a sample), as those with lower abundance produce low signal/noise ratios and no clear diagnostic ion fragments. The glycobiology field would benefit greatly by additional tools for quantitative measures of glycans, especially those of low abundance. As described in the complementary manuscript, we developed such a tool using the fluorescent linker F-MAPA to measure the abundance of all N-glycans and are in the process of developing techniques to quantitatively assess low abundance structures. We have added a sentence to the discussion highlighting the important consideration between quantitative and qualitative measures in glycobiology (Line 481-484).

They should state early on and in the results how many mice were used in the study.

We added the target number of mice for our study (Line 95-96) and ensured that each figure legend indicates the number of samples used or if it is a representative qualitative analysis from a single sample.

The authors mentioned in the introduction that so far no brain region specific analysis has been performed. However, there was a reference recently published that comprehensively covers brain glycosylation (Spatial and temporal diversity of glycome expression in mammalian brain, PNAS USA 117, 28743–28753 (2020)). Glycans were profiled from nine different brain regions in mouse and glycome diversity were compared. This should be properly described in the manuscript.

We do reference this PNAS paper in the introduction and have change the word "and" to "or" (Line 82) to account for the authors relying solely on a single analytical technique (LC-MS). Although they mention using MS/MS to confirm their structural identification of isomers, no data is shown or provided publicly. We report several compositions confirmed by MS/MS which are in conflict with their results, for example LacdiNAc structures, as emphasized in Fig. S2.

We expanded the sentence referencing this paper in discussion to include its study of different brain regions and during development (Line 385-387). We chose to not discuss specific discrepancies between our results in detail though include a few examples (Line 477-481), and

rather highlight the confidence in our structural assignments as supported by multiple analytic methods.

We have also added references to two recent papers that we have recently become aware of, both of which are limited by studying only N-glycans using a single analytical method of either 3 regions (Barboza *et al.*, Mol Cell Proteomics, 2021, ref # 44) or a sample size of 3 whole brains (Shen *et al.*, Nat. Methods, 2021, ref # 48).

Reviewer #2 (Remarks to the Author):

1. Technical detail. Were the brains perfused to remove blood before dissection? If not, the authors should comment on the extent to which the glycomic profiles are influenced by blood elements.

This is an important observation we have considered in detail but omitted in our initial submission. In brief, the brains are not perfused (added to line 647), quickly removed the skull, washed with PBS, dissected into separate regions, and snap frozen in liquid N₂.

We are confident that the contribution of blood elements to the brain glycan signal is minimal for several reasons, most significantly the lack of N-glycolylneuraminic acid (NeuGc) the brain. In mice, the majority of plasma N-glycans are sialylated (>90%) and almost exclusively by NeuGc at a ratio of ~70:1. It has been well established that brain tissue contains exclusively NeuAc, as the gene for the enzyme which converts NeuAc to NeuGc, cytidine monophospho-N-acetylneuraminic acid hydroxylase (*Cmah*) is not expressed in the brain of most animals (reviewed by Suzuki, 2006, reference # 52, also see Yamakawa *et al.*, 2018, *Nat. Comm.*). We did not detect *Cmah* expression in our brain RNAseq analysis, and nearly all of the sialylated glycans we detect in the brain (both N- and O-) are of the NeuAc form. In addition, the most abundant plasma N-glycan, the biantennary disialylated NeuGc structure (A2G2S2, m/z 2852), is not detected at a level above the signal/noise threshold in our brain samples. It is possible that non-plasma glycans from cellular components can influence the brain glycome, but again these would be expected to contain some fraction of NeuGc. We have added a sentence to summarize this information on N-glycans (Line 123-125) and O-glycans (214-218) in addition to the existing discussion.

2. It is assumed by this reviewer, but not stated anywhere, that the masses presented in the various tables, figures, and supplemental material are of the permethylated, perhaps sodiated, forms of the glycans. Please provide this information early in the manuscript.

Yes, all glycans analyzed by MS and MS/MS are presented as their permethylated (mono-sodiated) masses. We have added this information to both the N- and O-glycan results sections (Line 97 and 184).

3. Fig. 2 and others. Where more than one possible glycan structure is indicated for a particular mass (usually bracketed together), it is not always clear whether both structural isomers were detected in MS/MS or both structural isomers are possible but the data does not distinguish between the two. Are the two possibilities indicate to be considered “this and this,” vs. “this or this?”

We appreciate clarifying this important issue. Glycans which are shown in brackets at the same mass indicate that both isomers have been ruled in via MS/MS, essentially that “this and this” contribute to the signal at this location. We have added a sentence to indicate this (Line 103-105) and updated the legends for Fig 2, 3, 4, 5, S1, S2, and S3 to clarify this point.

4. Fig 2, panel C. What does “mono-antennary” mean? Is this different from hybrid? If so, how? Are there more than one “mono-antennary” glycans in the glycan set used for the heat-map? Pauci-mannose with extension on the 3-arm?

Our definition of glycan categories is based on the common nomenclature found in the Essentials of Glycobiology textbook (3rd Edition). Mono-antennary refers to glycans with one GlcNAc attached to the core α -1,3 Man, including 33 distinct glycan masses in our dataset and heat maps (Table S2) and composes between 13-19% of brain N-glycan abundance (Table 1). Hybrid glycans have at least one GlcNAc antenna attached to the core α -1,3 Man in addition to the extended Man residues on the α -1,6 Man arm. Pauci-mannose glycans refer to small oligo-mannose species (Man-2, Man-3, and Man-4) and are separate from high mannose (Man-5 through Man-9). To clarify this, we have added definitions for each N-glycan category to the beginning of the Supplementary Text and refer to it in the main text (Line 134-135).

5. Fig. 3, panel B. The justification for using the “#” placeholder is not clear. What happened to the signal intensity associated with the F2A1G1BH4 glycan? Also, for this entire figure, it is not clear what the values on the y-axis (x105 au) are meant to represent. This experiment is, I believe, a MALDI analysis of permethylated glycans released by the indicated digestions. It would be useful to provide some text that helps the reader understand how to compare the disparities in signal strengths across the three panels.

We have updated the figure legend to more clearly describe the need for the placeholder and what happened to the signal intensity from F2A1G1BH4 (Line 571-575). In brief, Endo H cleavage is between the two GlcNAc residues of the chitobiose core, whereas PNGase F cleaves between the asparagine and the N-glycan. Endo H cleavage of both FA1G1BH4 and F2A1G1BH4 generate the same fragment (FA1G1BH4) which are indistinguishable. Thus, we have used the # as a placeholder to align each parent mass across columns. In addition, we have added an explanation of y-axis to the figure legend, which represents the raw signal intensity (in arbitrary units) obtained from the MALDI-MS scaled to allow simple visual comparison between the panels (Line 565-568).

6. Fig. 5, panel D. The figure legend describing panel D is missing some words or a phrase.

We have corrected the error and added the word “cortex” (Line 603).

7. Fig. 7, panel E. The Fut8 reaction is over-simplified for presentation. It is not likely that Fut8 adds Fuc to the paucimannose glycan directly. Rather, the Fuc was probably added to a precursor that has been degraded to the paucimannose form. Would suggest that the MGAT1/GnT1 product be used for this illustration.

We have changed the image to include the MGAT1 product being fucosylated by Fut8.

8. The top 10 N-glycans (of 94) were presented in Fig. 2 and the top 15 in Fig. 3. Sorting the glycans for abundance in Table S1, it looks like the m/z values at 1345 and 2838, which were excluded from the top 10, are more highly abundant than at m/z=2244, which was included. Further clarification of why certain glycans were included for some analysis and excluded for others would be useful.

For Fig. 2, the top 10 glycans averaged across all four regions are shown. The m/z values of 1345 and 2838 are the 11th and 12th most abundant in the average which is why they are excluded in Fig. 2. I have added text to the figure legend to clarify (Line 550-551) and have added a column to Table S1 for the averages across the 4 regions.

In Fig. 3, we included the 15 most abundant N-glycans from cortex, including several structures (11th-15th) which are hybrids, pauci-mannose, or mixed. Their sensitivity to Endo H vs PNGase F highlights the unique structures present in the brain which have often been misclassified, for example the m/z 2070, which in plasma is A2G2, while in brain it is clearly A1BH5 given its unique sensitivity to Endo H. We have also confirmed m/z 2070 confirmed via MS/MS but opted to not include this data out of concern that having too much MS/MS data.

The authors mention that all of the glycans (assuming all of those in Table S1) were detected and quantifiable with a signal to noise of at least 6 to 1. Did the authors aggregate the features (number of antennae, core fucosylation, complex, hybrid, etc.) across all of the glycans, not just the top 10 or 15? If so, does a broader aggregation identify more subtle regional diversity? Same for O-linked glycans?

Yes, all of the quantitative comparisons between regions represent the aggregate signal of all 95 N- and 26 O-glycans based on the classification provided in Tables S2 and S3, with precisely the goal of detecting more subtle regional differences. For example, as shown in Table 1, the pauci-mannose N-glycans (which includes F-Man-3 at m/z 1345) are significantly different across regions ($p = 0.00005$), even though none are in the top 10 most abundant glycans when averaged across all 4 regions.

9. In the discussion, p. 20 beginning with line 446, the authors indicate that this study adds value on top of previous studies because it clarifies that “several prior assumptions on the identity, composition, and linkage of glycans in the brain were incorrect....” However, no reference to these previously incorrect assignments are provided, neither are specific examples cited. This claim should be elaborated upon or removed.

We have added a sentence elaborating on two of these specific claims and included references, in addition to a qualifying statement regarding the interpretation of results across techniques (Line 477-481).

10. The authors mention that data will be made available upon “reasonable request.” Have the authors considered submitting their raw MS and MS/MS data to an established repository, such as GlycoPost? Whether it is required by this journal or not, deposition to such a repository would be a very valuable contribution to the field. If this dataset, as the authors propose, will provide essential baseline data for future studies, it seems even more important for the raw data to be broadly accessible for future mining and interpretation. Likewise, it would be equally useful for the glycan compositions described in Table S1 (and in other places in the manuscript) to be associated with appropriate database accession numbers (GlyYouCan IDs). This association will greatly facilitate the uptake and dissemination of this data by various international glycan databases and knowledgebases. Again, I am unaware of whether this is required by the journal, but it would be a significant service to the field and a valuable addition to the authors’ dataset.

This is an important aspect of disseminating the data, in particular the raw data beyond our summary results available in the supplementary material. We have uploaded all of our raw data to public databases including GlycoPOST for glycomics data (GPST000213, available 9/21/22)

and GEO for RNAseq (GSE184516, available 9/24/21), and provided the accession numbers in the manuscript.

Reviewer #3 (Remarks to the Author):

- Fig 2 c– the p- values? In the Anova section only one star is shown in the figure whereas up to 3 stars are seen in Figure legend. Heat map seems to indicate that at least paucimannose, tetra antennary , sialylation should be significantly different between cortex and cerebellum?

The method for presenting asterisks and their corresponding p-value thresholds are different between panels B and C, with the number of asterisks (*, **, and ***) corresponding to the different thresholds in B, while the size of the asterisk corresponds with the threshold for the heatmaps. We added the full scale for the ANOVA significance threshold to the figure for clarification and to maintain consistency with the presentation style for O-glycans in Fig. 4.

We also noted that the asterisks in the heat map for N-glycans had some errors and have now corrected the placement to correspond with the correct values as shown in Table 1, which includes the significant difference of pauci-mannose structures between regions. Sialylation and tetra-antennary structures do show large percent changes on average between regions as indicated by the heatmap but fall short of significance as shown in Table 1 likely due to their relatively low abundance and larger variation (SEM) in the samples.

- Fig 3 nomenclature used should be defined e.g FA1B etc

We have added a key describing the nomenclature to the figure legend (Line 564-565), and additional details for each structure are available in Supplementary Table 2.

- What is the definition of the paucimannose structures seen? Essentials gives as M3-4 GlcNac2, CHEBI gives M1-4Fuc0-1GlcNac2 M5-9 would be considered oligomannose. The only putative pauci shown FM3 is not visible in the spectra and the other smaller varieties would also not be seen.

We have described pauci-mannose structures as Man-2, Man-3, and Man-4 with no terminal branching or modifications aside from core fucose, as according to Table S2. Man-5 through Man-9 are oligo-mannose and not included. We annotated all glycan masses above m/z of 1000 and with a signal to noise ratio greater than 6. We have added a description of pauci-mannose and other N-glycan characteristics used in our study to the Supplemental Text.

- Low sialic acid on N-glycans – was there a control for detection of sialylation after permethylation/MALDI?

Yes, we include a batch standard of serum or plasma during the purification to control for several factors including enzyme (PNGase F) activity, permethylation, and loss of specific structures such as sialic acid. We have also performed several controls using neuraminidase inhibitors during sample lysis/preparation to ensure there are not simply high levels of activity in the brain and see no difference. In addition, in the complementary paper submitted with this manuscript (Mealer *et al.*), we show that mice expressing a schizophrenia risk have impaired glycosylation including decreased sialic acid. We perform several additional experiments to assay N-glycans and sialic acid without permethylation and MALDI (fluorescent glycan derivatization using F-MAPA and sialic acid quantification using a commercially available kit) and find consistent results across the methods. In addition, as noted below, the robust level of sialylation on O-glycans which are undergoing all of the same steps in addition to beta-

elimination, are consistent with the compatibility of our methods to detect sialic acid when present, which was surprisingly low on N-glycans.

- “Simple linear regression of fucosylated vs sialylated O-glycans showed a strong and highly significant negative correlation in both O-GalNAc and O-mannose glycans, suggesting competition between these modifications in the brain” - Inverse correlation does not necessarily be explained by competition?

We agree, there are other possible interpretations of this observation. We have deleted the phrase about the interpretation relating to competition in the text and supplementary material (Line 202).

- Not clear if figs 2 and 3 and 4 are mouse or human brain derived – it is indicated neither in text or figure legend ? This needs to be made clearer throughout first sections? And were they the same sex?

We have added this info to the figure legend for Fig. 2, 3, 4, 5, 6, and 7 for consistency in addition to adding “male” in the main text at the beginning of the results section (Line 96). Data comparing sexes are shown in Fig. 5.

- O-glycans were highly sialylated – N glycans were not – were there specific sialyltransferase transcriptome expression differences?

Several sialyltransferases were differentially expressed between cortex and cerebellum, as summarized in Fig. 7 and presented in Table S7. We do note that *St3Gal2* and *St6galnac6* are some of the highest expressed sialyltransferases in the brain and have added a sentence to the discussion highlighting this point, as it may contribute to the difference between N- and O-glycan levels of sialylation (Line 456-461).

- How is presence of NeuGc explained on the O-glycans when you state for the NeuGc absence on the N-glycans “and the lack of expression of the enzyme which converts NeuAc to NeuGc in the brain”?

This issue is discussed in line 204-214 and we have added a sentence to clarify this observation (Line 214-218).

- Line 215 – this is not shown in Fig 3C? Do you mean Fig 4C?

Yes, we have corrected this error (Line 197).

- Fig 5 – why are there different n= for the N-linked and O-linked analyses?

During the purification of N- and O-glycans (dialysis of lipids, trypsinization, removal of N-glycans with PNGase F, B-elimination of O-glycans, purification, permethylation, and isolation for MS analysis), some samples do not produce any detectable signal for MS, presumably as the product is lost during isolation due to a technical issue. Although experiments were initiated with comparable numbers between region and sex, the final analysis is performed on samples that generate a measurable signal, which is the reason for different N in some experiments, particularly with the O-glycans.

- Fig 6: SNA binding is interesting as seems to indicated O-linked sialylation on a wide

range of proteins?

We agree and have included additional mention of this in the text and added a reference to our complementary manuscript with data which also supports this finding (Line 313-316).

- Would the MS method see mono- O-monosaccharides?

We do not detect mono-saccharides using our method, likely as they are of low relative abundance and primarily studied using lectin enrichment in most cases. This is mentioned in the discussion as several interesting monosaccharides have been described in the brain (Line 449-454).

- Have you considered polysialylation of the O-glycans? What is the "low abundance (of PSA) in the adult brain" in quantitative terms?

We observed several masses which likely correspond to a polysialylated O-glycan, including *m/z* 1460 and 1617. Previous studies using western blot have reported higher relative levels of PSA-NCAM in the developing brain vs adults, and we have included these references (115-118).

- "We highlight the value of complementary analyses as several prior assumptions on the identity, composition, and linkage of glycans in the brain were incorrect when relying on a single method" – would be good to give an example here.

We have added a sentence elaborating on two specific claims and included references, in addition to a qualifying statement regarding the interpretation of results across techniques (Line 477-484).

Methods:**- What Triton X was used for lysis?**

Triton X-100. We have updated the methods to include this information (Line 667).

- Where were human brain sections sourced?

Human Brain Cerebral Cortex Whole Tissue Lysate was purchased from Novus Biologicals (#NB820-59182), with 1mg used for glycomic analysis as indicated in the Methods (Line 653-655).

REVIEWERS' COMMENTS

Reviewer #1 (Remarks to the Author):

In this manuscript, the authors explored the unique properties of glycosylation in the mammalian brain using a variety of methods. This is exactly the type of research that needs to be done to understand the biochemistry of the brain, and the results are very interesting. The authors also supplemented the manuscript based on the comments mentioned in previous reviews.

Nevertheless, there are concerns about the gender-specific brain O-glycan outcomes inferred from the small sample analysis. Numerous human blood samples have been analyzed so far, but the understanding of the link between gender and glycosylation remains unclear. In this study, gender differences in brain O-glycans were investigated using only five mouse samples (male:3, female: 2). The authors noted that only mice with detected glycans were included in the results, as they could not detect brain glycans present in very small amounts due to sample loss caused in many experimental steps. Even if the circumstances are understood, further sample analysis should be considered to avoid overgeneralization of the results and to determine the reproducibility of the method and the reliability of the results. Also, since glycans were analyzed with MALDI-TOF MS, which is mainly used for qualitative analysis, there is a limit to guarantee quantitative results without internal standards or isotope labeling.

Reviewer #2 (Remarks to the Author):

The authors have effectively addressed the comments and concerns that I raised in my initial review.

Kind of a pain-in-the-neck that the number lines in the revised manuscript don't agree with the number lines in the response to the reviewers, though.

Nice paper.

Reviewer #3 (Remarks to the Author):

I believe that the authors have responded to all the reviewer's questions satisfactorily and altered the manuscript accordingly such that it is now suitable for publication.

The only question I still have is in relation to the "complementary" paper which is cited a few times in the text but is unpublished as a peer reviewed paper as yet.

Response to REVIEWERS' COMMENTS (NCOMMS-21-14663)

Reviewer #1 (Remarks to the Author):

In this manuscript, the authors explored the unique properties of glycosylation in the mammalian brain using a variety of methods. This is exactly the type of research that needs to be done to understand the biochemistry of the brain, and the results are very interesting. The authors also supplemented the manuscript based on the comments mentioned in previous reviews.

Nevertheless, there are concerns about the gender-specific brain O-glycan outcomes inferred from the small sample analysis. Numerous human blood samples have been analyzed so far, but the understanding of the link between gender and glycosylation remains unclear. In this study, gender differences in brain O-glycans were investigated using only five mouse samples (male:3, female: 2). The authors noted that only mice with detected glycans were included in the results, as they could not detect brain glycans present in very small amounts due to sample loss caused in many experimental steps. Even if the circumstances are understood, further sample analysis should be considered to avoid overgeneralization of the results and to determine the reproducibility of the method and the reliability of the results. Also, since glycans were analyzed with MALDI-TOF MS, which is mainly used for qualitative analysis, there is a limit to guarantee quantitative results without internal standards or isotope labeling.

We have tempered the conclusions and provided a new supplementary note to add some clarification and modified the discussion in the primary text. We have also added additional discussion to highlight the importance of quantitative studies.

Reviewer #2 (Remarks to the Author):

The authors have effectively addressed the comments and concerns that I raised in my initial review.

Kind of a pain-in-the-neck that the number lines in the revised manuscript don't agree with the number lines in the response to the reviewers, though.

Nice paper.

We apologize for the inconvenience caused by errors in the line numbers. The revised document did not format correctly in MS Word with changes tracked, likely resulting in the line number discrepancies.

Reviewer #3 (Remarks to the Author):

I believe that the authors have responded to all the reviewer's questions satisfactorily and altered the manuscript accordingly such that it is now suitable for publication.

The only question I still have is in relation to the "complementary" paper which is cited a few times in the text but is unpublished as a peer reviewed paper as yet.

This referenced work remains under review/revisions at *Molecular Psychiatry*. A copy of that manuscript was included in our initial submission and remains available on BioRxiv.